# SAM AS AN OPTIMAL RELAXATION OF BAYES

**Thomas Möllenhoff & Mohammad Emtiyaz Khan**
RIKEN Center for Advanced Intelligence Project
Tokyo, Japan
`{thomas.moellenhoff,emtiyaz.khan}@riken.jp`

## ABSTRACT

Sharpness-aware minimization (SAM) and related adversarial deep-learning methods can drastically improve generalization, but their underlying mechanisms are not yet fully understood. Here, we establish SAM as a relaxation of the Bayes objective where the expected negative-loss is replaced by the optimal convex lower bound, obtained by using the so-called Fenchel biconjugate. The connection enables a new Adam-like extension of SAM to automatically obtain reasonable uncertainty estimates, while sometimes also improving its accuracy. By connecting adversarial and Bayesian methods, our work opens a new path to robustness.

## 1 INTRODUCTION

Sharpness-aware minimization (SAM) (Foret et al., 2021) and related adversarial methods (Zheng et al., 2021; Wu et al., 2020; Kim et al., 2022) have been shown to improve generalization, calibration, and robustness in various applications of deep learning (Chen et al., 2022; Bahri et al., 2022), but the reasons behind their success are not fully understood. The original proposal of SAM was geared towards biasing training trajectories towards flat minima, and the effectiveness of such minima has various Bayesian explanations, for example, those relying on the optimization of description lengths (Hinton & Van Camp, 1993; Hochreiter & Schmidhuber, 1997), PAC-Bayes bounds (Dziugaite & Roy, 2017; 2018; Jiang et al., 2020; Alquier, 2021), or marginal likelihoods (Smith & Le, 2018). However, SAM is not known to directly optimize any such Bayesian criteria, even though some connections to PAC-Bayes do exist (Foret et al., 2021). The issue is that the 'max-loss' used in SAM fundamentally departs from a Bayesian-style 'expected-loss' under the posterior; see Fig. 1(a). The two methodologies are distinct, and little is known about their relationship.

Here, we establish a connection by using a relaxation of the Bayes objective where the expected negative-loss is replaced by the tightest convex lower bound. The bound is optimal, and obtained by Fenchel biconjugates which naturally yield the maximum loss used in SAM (Fig. 1(a)). From this, SAM can be seen as optimizing the relaxation of Bayes to find the mean of an isotropic Gaussian posterior while keeping the variance fixed. Higher variances lead to smoother objectives, which biases the solution towards flatter regions (Fig. 1(b)). Essentially, the result connects SAM and Bayes through a Fenchel biconjugate that replaces the expected loss in Bayes by a maximum loss.

What do we gain by this connection? The generality of our result makes it possible to easily combine the complementary strengths of SAM and Bayes. For example, we show that the relaxed-Bayes objective can be used to learn the variance parameter, which yields an Adam-like extension of SAM (Alg. 1). The variances are cheaply obtained from the vector that adapts the learning rate, and SAM's hyperparameter is adjusted for each parameter dimension via the variance vector. The extension improves the performance of SAM on standard benchmarks while giving comparable uncertainty estimates to the best Bayesian methods.

Our work complements similar extensions for SGD, RMSprop, and Adam from the Bayesian deep-learning community (Gal & Ghahramani, 2016; Mandt et al., 2017; Khan et al., 2018; Osawa et al., 2019). So far there is no work on such connections between SAM and Bayes, except for a recent empirical study by Kaddour et al. (2022). Husain & Knoblauch (2022) give an adversarial interpretation of Bayes, but it does not cover methods like SAM. Our work here is focused on connections to approximate Bayesian methods, which can have both theoretical and practical impact. We discuss a new path to robustness by connecting the two fields of adversarial and Bayesian methodologies.

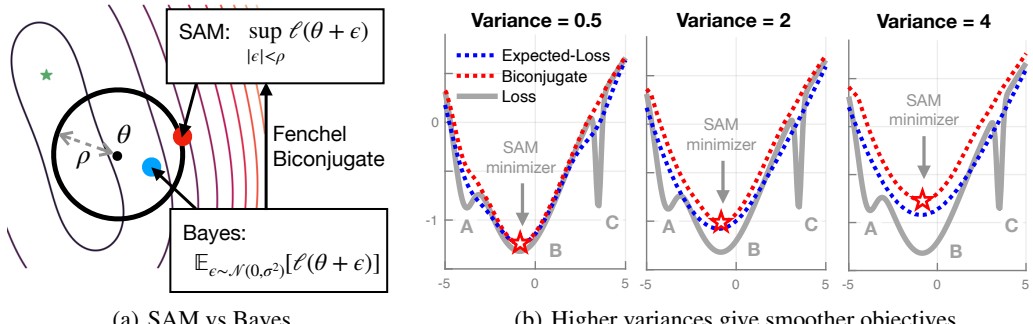

(a) SAM vs Bayes
(b) Higher variances give smoother objectives

Figure 1: Panel (a) highlights the main difference between SAM and Bayes. SAM uses max-loss (the red dot), while Bayes uses expected-loss (the blue dot shows $\ell(\theta + \epsilon)$ for an $\epsilon \sim \mathcal{N}(\epsilon \,|\, 0, \sigma^2)$). Our main result in Theorem 2 connects the two by using the optimal convex relaxation (Fenchel biconjugate) of the negative expected loss. It shows that the role of $\rho$ and $\sigma$ are exactly the same, and a SAM minimizer obtained for a fixed $\rho$ can always be recovered from the relaxation for some $\sigma$. An example is shown in Panel (b) for a loss (gray line) with 3 local minima indicated by A, B, and C. The expected loss is smoother (blue line) but the relaxation, which upper bounds it, is even more smooth (red line). Higher $\sigma$ give smoother objectives where sharp minima A and C slowly disappear. The SAM minimizer is shown with a red star which matches the minimizer of the relaxation.

## 2 SAM AS AN OPTIMAL RELAXATION OF BAYES

In SAM, the training loss $\ell(\boldsymbol{\theta})$ is replaced by the maximum loss within a ball of radius $\rho > 0$ around the parameter $\boldsymbol{\theta} \in \Theta \subset \mathbb{R}^P$, as shown below for an $\ell_2$-regularized problem,

$$\mathcal{E}_{\text{SAM}}(\boldsymbol{\theta}; \rho, \delta) = \sup_{\|\boldsymbol{\epsilon}\| \leq \rho} \ell(\boldsymbol{\theta} + \boldsymbol{\epsilon}) + \frac{\delta}{2}\|\boldsymbol{\theta}\|^2, \tag{1}$$

with $\delta > 0$ as the regularization parameter. The use of 'maximum' above differs from Bayesian strategies that use 'expectation', for example, the variational formulation by Zellner (1988),

$$\mathcal{L}(q) = \mathbb{E}_{\boldsymbol{\theta} \sim q}\left[-\ell(\boldsymbol{\theta})\right] - \mathbb{D}_{\text{KL}}(q(\boldsymbol{\theta}) \,\|\, p(\boldsymbol{\theta})), \tag{2}$$

where $q(\boldsymbol{\theta})$ is the generalized posterior (Zhang, 1999; Catoni, 2007), $p(\boldsymbol{\theta})$ is the prior, and $\mathbb{D}_{\text{KL}}(\cdot \,\|\, \cdot)$ is the Kullback-Leibler divergence (KLD). For an isotropic Gaussian posterior $q(\boldsymbol{\theta}) = \mathcal{N}(\boldsymbol{\theta} \,|\, \mathbf{m}, \sigma^2 \mathbf{I})$ and prior $p(\boldsymbol{\theta}) = \mathcal{N}(\boldsymbol{\theta} \,|\, 0, \mathbf{I}/\delta)$, the objective with respect to the mean $\mathbf{m}$ becomes

$$\mathcal{E}_{\text{Bayes}}(\mathbf{m}; \sigma, \delta) = \mathbb{E}_{\boldsymbol{\epsilon}' \sim \mathcal{N}(0, \sigma^2 \mathbf{I})}\left[\ell(\mathbf{m} + \boldsymbol{\epsilon}')\right] + \frac{\delta}{2}\|\mathbf{m}\|^2, \tag{3}$$

which closely resembles Eq. 1, but with the maximum replaced by an expectation. The above objective is obtained by simply plugging in the posterior and prior, then collecting the terms that depend on $\mathbf{m}$, and finally rewriting $\boldsymbol{\theta} = \mathbf{m} + \boldsymbol{\epsilon}'$.

There are some resemblances between the two objectives which indicate that they might be connected. For example, both use local neighborhoods: the isotropic Gaussian can be seen as a 'softer' version of the 'hard-constrained' ball used in SAM. The size of the neighborhood is decided by their respective parameters ($\sigma$ or $\rho$ in Fig. 1(a)). But, apart from these resemblances, are there any deeper connections? Here, we answer this question.

We will show that Eq. 1 is equivalent to optimizing with respect to $\mathbf{m}$ an *optimal* convex relaxation of Eq. 3 while keeping $\sigma$ fixed. See Theorem 2 where an equivalence is drawn between $\sigma$ and $\rho$, revealing similarities between the mechanisms used in SAM and Bayes to favor flatter regions. Fig. 1(b) shows an illustration where the relaxation is shown to upper bound the expected loss. We will now describe the construction of the relaxation based on Fenchel biconjugates.

### 2.1 FROM EXPECTED-LOSS TO MAX-LOSS VIA FENCHEL BICONJUGATES

We use results from convex duality to connect expected-loss to max-loss. Our main result relies on Fenchel conjugates (Hiriart-Urruty & Lemaréchal, 2001), also known as convex conjugates. Con-

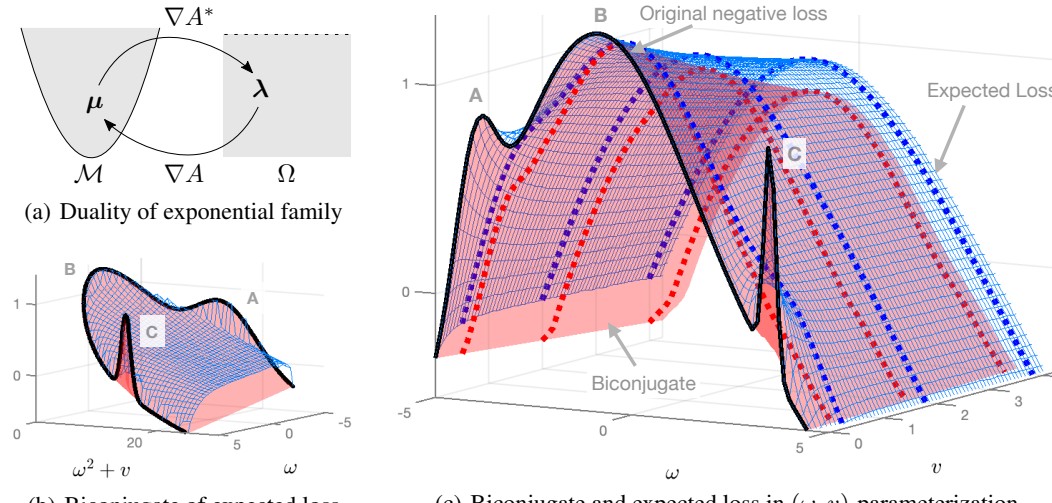

(a) Duality of exponential family

(b) Biconjugate of expected loss

(c) Biconjugate and expected loss in $(\omega, v)$-parameterization

Figure 2: Panel (a) shows the dual-structure of exponential-family for a Gaussian defined in Eq. 6. The space $\mathcal{M}$ lies above the parabola, while $\Omega$ is a half space. Panel (b) shows $f(\boldsymbol{\mu})$ and its biconjugate $f^{**}(\boldsymbol{\mu})$ with the blue mesh and red surface respectively. We use the loss from Fig. 1(b). The $f^{**}$ function perfectly matches the original loss at the boundary where $v \to 0$ (black curve), clearly showing the regions for three minima A, B and C. Panel (c) shows the same but with respect to the standard mean-variance $(\omega, v)$ parameterization. Here, convexity of $f^{**}$ is lost, but we see more clearly the negative $\ell(\omega)$ appear at $v \to 0$ (black curve). At this limit, we also get $f(\boldsymbol{\mu}) = f^{**}(\boldsymbol{\mu}) = -\ell(\omega)$, but for higher variances, both $f$ and $f^{**}$ are smoothed version of $\ell(\omega)$. The slices at $v = 0.5, 2$, and 4 give rise to the curves shown in Fig. 1(b).

sider a function $f(\mathbf{u})$, defined over $\mathbf{u} \in \mathcal{U}$ and taking values in the extended real line. Given a dual space $\mathbf{v} \in \mathcal{V}$, the Fenchel biconjugate $f^{**}$ and conjugate $f^*$ are defined as follows,

$$f^{**}(\mathbf{u}) = \sup_{\mathbf{v}' \in \mathcal{V}} \langle \mathbf{v}', \mathbf{u} \rangle - f^*(\mathbf{v}'), \qquad f^*(\mathbf{v}') = \sup_{\mathbf{u}'' \in \mathcal{U}} \langle \mathbf{u}'', \mathbf{v}' \rangle - f(\mathbf{u}''), \qquad (4)$$

where $\langle \cdot, \cdot \rangle$ denotes the inner product. Note that $f$ need not be convex, but both $f^*$ and $f^{**}$ are convex functions. Moreover, the biconjugate is the *optimal* convex lower bound: $f(\mathbf{u}) \geq f^{**}(\mathbf{u})$.

We will use the conjugate functions to connect the max-loss in SAM to the expected-loss involving regular, minimal exponential-family distributions (Wainwright et al., 2008) of the following form[1],

$$q_{\boldsymbol{\lambda}}(\boldsymbol{\theta}) = \exp\left( \langle \boldsymbol{\lambda}, \mathbf{T}(\boldsymbol{\theta}) \rangle - A(\boldsymbol{\lambda}) \right), \qquad (5)$$

with natural parameter $\boldsymbol{\lambda} \in \Omega$, sufficient statistics $\mathbf{T}(\boldsymbol{\theta})$, log-partition function $A(\boldsymbol{\lambda})$, and a base measure of 1 with $\Omega = \{\boldsymbol{\lambda} : A(\boldsymbol{\lambda}) < \infty\}$. Dual spaces naturally arise for such distributions and can be used to define conjugates. For every $\boldsymbol{\lambda} \in \Omega$, there exists a unique 'dual' parameterization in terms of the expectation parameter $\boldsymbol{\mu} = \mathbb{E}_{\boldsymbol{\theta} \sim q_{\boldsymbol{\lambda}}}[\mathbf{T}(\boldsymbol{\theta})]$. The space of all valid $\boldsymbol{\mu}$ satisfies $\boldsymbol{\mu} = \nabla A(\boldsymbol{\lambda})$, and all such $\boldsymbol{\mu}$ form the interior of a dual space which we denote by $\mathcal{M}$. We can also write $\boldsymbol{\lambda}$ in terms of $\boldsymbol{\mu}$ using the inverse $\boldsymbol{\lambda} = \nabla A^*(\boldsymbol{\mu})$, and in general, both $\boldsymbol{\mu}$ and $\boldsymbol{\lambda}$ are valid parameterizations.

In what follows, we will always use $\boldsymbol{\mu}$ to parameterize the distribution, denoting $q_{\boldsymbol{\mu}}(\boldsymbol{\theta})$. As an example, for a Gaussian $\mathcal{N}(\theta \,|\, \omega, v)$ over scalar $\theta$ with mean $\omega$ and variance $v > 0$, we have,

$$\mathbf{T}(\theta) = (\theta, \theta^2), \quad \boldsymbol{\mu} = \left( \omega, \omega^2 + v \right), \quad \boldsymbol{\lambda} = \left( \omega/v, -1/(2v) \right). \qquad (6)$$

Because $v > 0$, the space $\mathcal{M}$ is the epigraph of parabola $(\omega, \omega^2)$, and $\Omega$ is a negative half space because the second natural parameter is constrained to be negative; see Fig. 2(a). The dual spaces can be used to define the Fenchel conjugates.

We will use the conjugates of the 'expected negative-loss' acting over $\boldsymbol{\mu} \in \text{interior}(\mathcal{M})$,

$$f(\boldsymbol{\mu}) = \mathbb{E}_{\boldsymbol{\theta} \sim q_{\boldsymbol{\mu}}} \left[ -\ell(\boldsymbol{\theta}) \right]. \qquad (7)$$

---

[1]The notation used in the paper is summarized in Table 10 in App. H.

For all $\boldsymbol{\mu} \notin \text{interior}(\mathcal{M})$, we set $f(\boldsymbol{\mu}) = +\infty$. Using Eq. 4, we can obtain the biconjugate,

$$f^{**}(\boldsymbol{\mu}) = \sup_{\boldsymbol{\lambda}' \in \Omega} \langle \boldsymbol{\lambda}', \boldsymbol{\mu} \rangle - \left[ \sup_{\boldsymbol{\mu}'' \in \mathcal{M}} \langle \boldsymbol{\mu}'', \boldsymbol{\lambda}' \rangle - \mathbb{E}_{\boldsymbol{\theta} \sim q_{\boldsymbol{\mu}''}} \left[ -\ell(\boldsymbol{\theta}) \right] \right], \tag{8}$$

where we use $\boldsymbol{\lambda}'$ and $\boldsymbol{\mu}''$ to denote arbitrary points from $\Omega$ and $\mathcal{M}$, respectively, and avoid confusion to the change of coordinates $\boldsymbol{\mu}' = \nabla A(\boldsymbol{\lambda}')$.

We show in App. A that, for an isotropic Gaussian, the biconjugate takes a form where max-loss automatically emerges. We define $\boldsymbol{\mu}$ to be the expectation parameter of a Gaussian $\mathcal{N}(\boldsymbol{\theta} \,|\, \boldsymbol{\omega}, v\mathbf{I})$, and $\boldsymbol{\lambda}'$ to be the natural parameter of a (different) Gaussian $\mathcal{N}(\boldsymbol{\theta} \,|\, \mathbf{m}, \sigma^2\mathbf{I})$. The equation below shows the biconjugate with the max-loss included in the second term,

$$f^{**}(\boldsymbol{\mu}) = \sup_{\mathbf{m},\sigma} \mathbb{E}_{\boldsymbol{\theta} \sim q_{\boldsymbol{\mu}}} \left[ -\frac{1}{2\sigma^2} \|\boldsymbol{\theta} - \mathbf{m}\|^2 \right] - \left[ \sup_{\boldsymbol{\epsilon}} \ell(\mathbf{m} + \boldsymbol{\epsilon}) - \frac{1}{2\sigma^2} \|\boldsymbol{\epsilon}\|^2 \right]. \tag{9}$$

The max-loss here is in a proximal form, which has an equivalence to the trust-region form used in Eq. 1 (Parikh & Boyd, 2013, Sec. 3.4). The first term in the biconjugate arises after simplification of the $\langle \boldsymbol{\lambda}', \boldsymbol{\mu} \rangle$ term in Eq. 8, while the second term is obtained by using the theorem below to switch the maximum over $\boldsymbol{\mu}''$ to a maximum over $\boldsymbol{\theta}$, and subsequently over $\boldsymbol{\epsilon}$ by rewriting $\boldsymbol{\theta} = \mathbf{m} + \boldsymbol{\epsilon}$.

**Theorem 1.** *For a Gaussian distribution that takes the form* (5)*, the conjugate $f^*(\boldsymbol{\lambda}')$ can be obtained by taking the supremum over $\Theta$ instead of $\mathcal{M}$, that is,*

$$\sup_{\boldsymbol{\mu}'' \in \mathcal{M}} \langle \boldsymbol{\mu}'', \boldsymbol{\lambda}' \rangle - \mathbb{E}_{\boldsymbol{\theta} \sim q_{\boldsymbol{\mu}''}} [-\ell(\boldsymbol{\theta})] = \sup_{\boldsymbol{\theta} \in \Theta} \langle \mathbf{T}(\boldsymbol{\theta}), \boldsymbol{\lambda}' \rangle - [-\ell(\boldsymbol{\theta})], \tag{10}$$

*assuming that $\ell(\boldsymbol{\theta})$ is lower-bounded, continuous and majorized by a quadratic function. Moreover, there exists $\boldsymbol{\lambda}'$ for which $f^*(\boldsymbol{\lambda}')$ is finite and, assuming $\ell(\boldsymbol{\theta})$ is coercive, $\text{dom}(f^*) \subseteq \Omega$.*

A proof of this theorem is in App. B, where we also discuss its validity for generic distributions.

The red surface in Fig. 2(b) shows the convex lower bound $f^{**}$ to the function $f$ (shown with the blue mesh) for a Gaussian with mean $\omega$ and variance $v$ (Eq. 6). Both functions have a parabolic shape which is due to the shape of $\mathcal{M}$. Fig. 2(c) shows the same but with respect to the standard mean-variance parameterization $(\omega, v)$. In this parameterization, convexity of $f^{**}$ is lost, but we see more clearly the negative $\ell(\omega)$ appear at $v \to 0$ (shown with the thick black curve). At the limit, we get $f(\boldsymbol{\mu}) = f^{**}(\boldsymbol{\mu}) = -\ell(\omega)$, but for higher variances, both $f$ and $f^{**}$ are smoothed versions of $\ell(\omega)$. Fig. 2(c) shows three pairs of blue and red lines. The blue line in each pair is a slice of the function $f$, while the red line is a slice of $f^{**}$. The values of $v$ for which the functions are plotted are $0.5, 2$, and $4$. The slices at these variances are visualized in Fig. 1(b) along with the original loss.

The function $f^{**}(\boldsymbol{\mu})$ is a 'lifted' version of the one-dimensional $\ell(\theta)$ to a two-dimensional $\boldsymbol{\mu}$-space. Such liftings have been used in the optimization literature (Bauermeister et al., 2022). Our work connects them to exponential families and probabilistic inference. The convex relaxation preserves the shape at the limit $v \to 0$, while giving smoothed lower bounds for all $v > 0$. For a high enough variance the local minima at A and C disappear, biasing the optimization towards flatter region B of the loss. We will now use this observation to connect to SAM.

## 2.2 SAM as an optimal relaxation of the Bayes objective

We will now show that a minimizer of the SAM objective in Eq. 1 can be recovered by minimizing a relaxation of the Bayes objective where expected-loss $f$ is replaced by its convex lower bound $f^{**}$. The new objective, which we call relaxed-Bayes, always lower bounds the Bayes objective of Eq. 2,

$$\sup_{\boldsymbol{\mu} \in \mathcal{M}} \mathcal{L}(q_{\boldsymbol{\mu}}) \geq \sup_{\boldsymbol{\mu} \in \mathcal{M}} f^{**}(\boldsymbol{\mu}) - \mathbb{D}_{\text{KL}}(q_{\boldsymbol{\mu}}(\boldsymbol{\theta}) \,\|\, p(\boldsymbol{\theta})) \tag{11}$$

$$= \sup_{\boldsymbol{\mu} \in \mathcal{M}} \left\{ \sup_{\mathbf{m},\sigma} - \left[ \sup_{\boldsymbol{\epsilon}} \ell(\mathbf{m} + \boldsymbol{\epsilon}) - \frac{1}{2\sigma^2} \|\boldsymbol{\epsilon}\|^2 \right] - \mathbb{D}_{\text{KL}}\left( q_{\boldsymbol{\mu}}(\boldsymbol{\theta}) \,\|\, \frac{1}{\mathcal{Z}} p(\boldsymbol{\theta}) e^{-\frac{1}{2\sigma^2}\|\boldsymbol{\theta} - \mathbf{m}\|^2} \right) + \log \mathcal{Z} \right\},$$

where in the second line we substitute $f^{**}(\boldsymbol{\mu})$ from (9), and merge $\mathbb{E}_{\boldsymbol{\theta} \sim q_{\boldsymbol{\mu}}} [\frac{1}{2\sigma^2} \|\boldsymbol{\theta} - \mathbf{m}\|^2]$ with the KLD term. $\mathcal{Z}$ denotes the normalizing constant of the second distribution in the KLD. The relaxed-Bayes objective has a specific difference-of-convex (DC) structure because both $f^{**}$ and KLD are

convex with respect to $\boldsymbol{\mu}$. Optimizing the relaxed-Bayes objective relates to double-loop algorithms where the inner-loop uses max-loss to optimize for $(\mathbf{m}, \sigma)$, while the outer loop solves for $\boldsymbol{\mu}$. We will use the structure to simplify and connect to SAM.

Due to the DC structure, it is possible to switch the order of maximization with respect to $\boldsymbol{\mu}$ and $(\mathbf{m}, \sigma)$, and eliminate $\boldsymbol{\mu}$ by a direct maximization (Toland, 1978). The maximum occurs when the KLD term is 0 and the two distributions in its arguments are equal. A detailed derivation is in App. C, which gives us a *minimization* problem over $(\mathbf{m}, \sigma)$

$$\mathcal{E}_{\text{relaxed}}(\mathbf{m}, \sigma; \delta') = \left[\sup_{\boldsymbol{\epsilon}} \ell(\mathbf{m} + \boldsymbol{\epsilon}) - \frac{1}{2\sigma^2}\|\boldsymbol{\epsilon}\|^2\right] + \underbrace{\frac{\delta'}{2}\|\mathbf{m}\|^2 - \frac{P}{2}\log(\sigma^2\delta')}_{=-\log\mathcal{Z}}, \qquad (12)$$

where $\delta' = 1/(\sigma^2 + 1/\delta_0)$ and we use a Gaussian prior $p(\boldsymbol{\theta}) = \mathcal{N}(\boldsymbol{\theta} \,|\, 0, \mathbf{I}/\delta_0)$. This objective uses max-loss in the proximal form, which is similar to the SAM loss in Eq. 1 with the difference that the hard-constraint is replaced by a softer penalty term.

The theorem below is our main result which shows that, for a given $\rho$, a minimizer of the SAM objective in Eq. 1 can be recovered by optimizing Eq. 12 with respect to $\mathbf{m}$ while keeping $\sigma$ fixed.

**Theorem 2.** *For every $(\rho, \delta)$, there exist $(\sigma, \delta')$ such that*

$$\arg\min_{\boldsymbol{\theta}\in\Theta} \mathcal{E}_{\text{SAM}}(\boldsymbol{\theta}; \rho, \delta) = \arg\min_{\mathbf{m}\in\Theta} \mathcal{E}_{\text{relaxed}}(\mathbf{m}, \sigma; \delta'), \qquad (13)$$

*assuming that the SAM-perturbation satisfies $\|\boldsymbol{\epsilon}\| = \rho$ at a stationary point.*

A proof is given in App. D and essentially follows from the equivalences between proximal and trust-region formulations. The result establishes SAM as a special case of relaxed-Bayes, and shows that $\rho$ and $\sigma$ play the exactly same role. This formalizes the intuitive resemblance discussed at the beginning of Sec. 2, and suggests that, in both Bayes and SAM, flatter regions are preferred for higher $\sigma$ because such values give rise to smoother objectives (Fig. 1(b)).

The result can be useful for combining complementary strengths of SAM and Bayes. For example, uncertainty estimation in SAM can now be performed by estimating $\sigma$ through the relaxed-Bayes objective. Another way is to exploit the DC structure in $\boldsymbol{\mu}$-space to devise new optimization procedures to improve SAM. In Sec. 3, we will demonstrate both of these improvements of SAM.

It might also be possible to use SAM to improve Bayes, something we would like to explore in the future. For Bayes, it still is challenging to get a good performance in deep learning while optimizing for both mean and variance (despite some recent success (Osawa et al., 2019)). Often, non-Bayesian methods are used to set the variance parameter (Plappert et al., 2018; Fortunato et al., 2018; Bisla et al., 2022). The reasons behind the bad performance remain unclear. It is possible that a simplistic posterior is to blame (Foong et al., 2020; Fortuin et al., 2022; Coker et al., 2022), but our result suggests to not disregard the opposite hypothesis (Farquhar et al., 2020). SAM works well, despite using a simple posterior, and it is possible that the poor performance of Bayes is due to poor algorithms, not simplistic posteriors. We believe that our result can be used to borrow algorithmic techniques from SAM to improve Bayes, and we hope to explore this in the future.

## 3 BAYESIAN-SAM TO OBTAIN UNCERTAINTY ESTIMATES FOR SAM

In this section, we will show an application of our result to improve an aspect of SAM. We will derive a new algorithm that can automatically obtain reasonable uncertainty estimates for SAM. We will do so by optimizing the relaxed-Bayes objective to get $\sigma$. A straightforward way is to directly optimize Eq. 12, but we will use the approach followed by Khan et al. (2018), based on the natural-gradient method of Khan & Rue (2021), called the Bayesian learning rule (BLR). This approach has been shown to work well on large problems, and is more promising because it leads to Adam-like algorithms, enabling us to leverage existing deep-learning techniques to improve training (Osawa et al., 2019), for example, by using momentum and data augmentation.

To learn variances on top of SAM, we will use a multivariate Gaussian posterior $q_{\boldsymbol{\mu}}(\boldsymbol{\theta}) = \mathcal{N}(\boldsymbol{\theta} \,|\, \boldsymbol{\omega}, \mathbf{V})$ with mean $\mathbf{m}$ and a diagonal covariance $\mathbf{V} = \text{diag}(\mathbf{s})^{-1}$, where $\mathbf{s}$ is a vector of precision values (inverse of the variances). The posterior is more expressive than the isotropic Gaussian

**Algorithm 1** Our Bayesian-SAM (bSAM) is a simple modification of SAM with Adam . Just add the red boxes, or use them to replace the blue boxes next to them. The advantage of bSAM is that it automatically obtains variances $\boldsymbol{\sigma}^2$ through the vector $\mathbf{s}$ that adapts the learning rate (see line 11). Two other main changes are in line 3 where Gaussian sampling is added, and line 5 where the vector $\mathbf{s}$ is used to adjust the perturbation $\boldsymbol{\epsilon}$. The change in line 8 improves the variance by adding the regularizer $\delta$ and a constant $\gamma$ (just like Adam), while line 9 uses a Newton-like update where $\mathbf{s}$ is used instead of $\sqrt{\mathbf{s}}$. We denote by $\mathbf{a} \cdot \mathbf{b}$, $\mathbf{a}/\mathbf{b}$ and $|\mathbf{a}|$, the element-wise multiplication, division and absolute value, respectively. $N$ denotes the number of training examples.

**Input:** Learning rate $\alpha, \beta_1$, $L_2$-regularizer $\delta$, SAM parameter $\rho$, $\beta_2 = 0.999$, $\gamma = $ $10^{-8}$ $0.1$

1: Initialize mean $\boldsymbol{\omega} \leftarrow$ (NN weight init), scale $\mathbf{s} \leftarrow$ $\mathbf{0}$ $\mathbf{1}$ , and momentum vector $\mathbf{g}_m \leftarrow \mathbf{0}$

2: **while** not converged **do**

3: $\quad \boldsymbol{\theta} \leftarrow \boldsymbol{\omega}$ $+ \mathbf{e}$, where $\mathbf{e} \sim \mathcal{N}(\mathbf{e}\,|\,0, \boldsymbol{\sigma}^2), \ \boldsymbol{\sigma}^2 \leftarrow \mathbf{1}/(N \cdot \mathbf{s})$ {Sample to improve the variance}

4: $\quad \mathbf{g} \leftarrow \frac{1}{B} \sum_{i \in \mathcal{M}} \nabla \ell_i(\boldsymbol{\theta})$ where $\mathcal{M}$ is minibatch of $B$ examples

5: $\quad \boldsymbol{\epsilon} \leftarrow \rho$ $\dfrac{\mathbf{g}}{\|\mathbf{g}\|}$ $\dfrac{\mathbf{g}}{\mathbf{s}}$ {Rescale by a vector $\mathbf{s}$, instead of a scalar $\|\mathbf{g}\|$ used in SAM}

6: $\quad \mathbf{g}_\epsilon \leftarrow \frac{1}{B} \sum_{i \in \mathcal{M}} \nabla \ell_i(\boldsymbol{\omega} + \boldsymbol{\epsilon})$

7: $\quad \mathbf{g}_m \leftarrow \beta_1 \mathbf{g}_m + (1 - \beta_1)(\mathbf{g}_\epsilon + \delta \boldsymbol{\omega})$

8: $\quad \mathbf{s} \leftarrow \beta_2 \mathbf{s} + (1 - \beta_2)$ $(\mathbf{g}_\epsilon + \delta \boldsymbol{\omega})^2$ $\left(\sqrt{\mathbf{s}} \cdot |\mathbf{g}| + \delta + \gamma\right)$ {Improved variance estimate}

9: $\quad \boldsymbol{\omega} \leftarrow \boldsymbol{\omega} - \alpha$ $\dfrac{\mathbf{g}_m}{\sqrt{\mathbf{s}} + \gamma}$ $\dfrac{\mathbf{g}_m}{\mathbf{s}}$ {Scale the gradient by $\mathbf{s}$ instead of $\sqrt{\mathbf{s}}$}

10: **end while**

11: Return the mean $\mathbf{m}$ and variance $\boldsymbol{\sigma}^2 \leftarrow \mathbf{1}/(N \cdot \mathbf{s})$

used in the previous section, and is expected to give better uncertainty estimates. As before, we will use a Gaussian prior $p(\boldsymbol{\theta}) = \mathcal{N}(\boldsymbol{\theta}\,|\,0, \mathbf{I}/\delta_0)$.

The BLR uses gradients with respect to $\boldsymbol{\mu}$ to update $\boldsymbol{\lambda}$. The pair $(\boldsymbol{\mu}, \boldsymbol{\lambda})$ is defined similarly to Eq. 6. We will use the update given in Khan & Rue (2021, Eq. 6) which when applied to Eq. 11 gives us,

$$\boldsymbol{\lambda} \leftarrow (1 - \alpha)\boldsymbol{\lambda} + \alpha \left[\nabla f^{**}(\boldsymbol{\mu}) + \boldsymbol{\lambda}_0\right], \tag{14}$$

where $\alpha > 0$ is the learning rate and $\boldsymbol{\lambda}_0 = (0, -\delta_0 \mathbf{I}/2)$ is the natural parameter of $p(\boldsymbol{\theta})$. This is obtained by using the fact that $\nabla_{\boldsymbol{\mu}} \mathbb{D}_{\mathrm{KL}}(q_{\boldsymbol{\mu}}(\boldsymbol{\theta}) \,\|\, p(\boldsymbol{\theta})) = \boldsymbol{\lambda} - \boldsymbol{\lambda}_0$; see Khan & Rue (2021, Eq. 23). For $\alpha = 1$, the update is equivalent to the difference-of-convex (DC) algorithm (Tao & An, 1997). The BLR generalizes this procedure by using $\alpha < 1$, and automatically exploits the DC structure in $\boldsymbol{\mu}$-space, unlike the naive direct minimization of (12) with gradient descent.

The update (14) requires $\nabla f^{**}$, which is obtained by maximizing a variant of Eq. 9, that replaces the isotropic covariance by a diagonal one. For a scalable method, we solve them inexactly by using the 'local-linearization' approximation used in SAM, along with an additional approximation to do single block-coordinate descent step to optimize for the variances. The rest of the derivations is similar to the previous applications of the BLR; see Khan et al. (2018); Osawa et al. (2019). The details are in App. E, and the resulting algorithm, which we call Bayesian-SAM is shown in Alg. 1.

bSAM differs slightly from a direct application of Adam (Kingma & Ba, 2015) on the SAM objective, but uses the same set of hyperparameters. The differences are highlighted in Alg. 1, where bSAM is obtained by replacing the blue boxes with the red boxes next to them. For SAM with Adam, we use the method discussed in Bahri et al. (2022); Kwon et al. (2021) for language models. The main change is in line 5, where the perturbation $\boldsymbol{\epsilon}$ is adapted by using a *vector* $\mathbf{s}$, as opposed to SAM which uses a *scalar* $\|\mathbf{g}\|$. Modifications in line 3 and 8 are aimed to improve the variance estimates because with just Adam we may not get good variances; see the discussion in Khan et al. (2018, Sec 3.4). Finally, the modification in line 9 is due to the Newton-like updates arising from BLR (Khan & Rue, 2021) where $\mathbf{s}$ is used instead of $\sqrt{\mathbf{s}}$ to scale the gradient as explained in App. E.

Due to their overall structural similarity, the bSAM algorithm has a similar computational complexity to Adam with SAM. The only additional overhead compared to SAM-type methods is the computation of a random noise perturbation which is negligible compared to gradient computations.

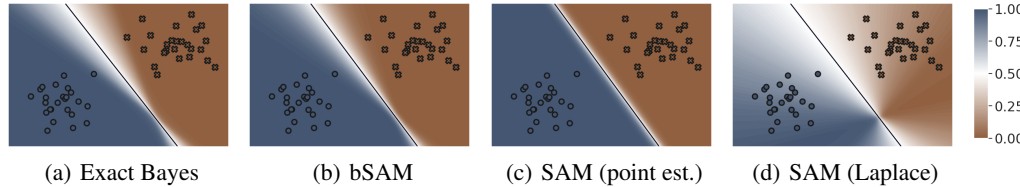

|  (a) Exact Bayes | (b) bSAM | (c) SAM (point est.) | (d) SAM (Laplace) |

Figure 3: For a 2D logistic regression, bSAM gives predictive uncertainties that are similar to those obtained by an *exact* Bayesian posterior. White areas show uncertain predictive probabilities around $0.5$. SAM's point estimate gives overconfident predictions, while Laplace leads to underconfidence.

| Model / Data | Method | Accuracy ↑ (higher is better) | NLL ↓ (lower is better) | ECE ↓ (lower is better) | AUROC ↑ (higher is better) |
|---|---|---|---|---|---|
| ResNet-20-FRN CIFAR-10 | SGD | $86.55_{(0.35)}$ | $0.56_{(0.014)}$ | $0.0839_{(0.003)}$ | $0.878_{(0.005)}$ |
| | SAM-SGD | $87.49_{(0.26)}$ | $0.49_{(0.019)}$ | $0.0710_{(0.003)}$ | $0.891_{(0.004)}$ |
| | SWAG | $86.80_{(0.10)}$ | $0.53_{(0.017)}$ | $0.0774_{(0.001)}$ | $0.880_{(0.006)}$ |
| | VOGN | $87.30_{(0.24)}$ | $0.38_{(0.004)}$ | $0.0315_{(0.003)}$ | $0.890_{(0.003)}$ |
| | Adam | $80.85_{(0.98)}$ | $0.83_{(0.063)}$ | $0.1317_{(0.011)}$ | $0.820_{(0.013)}$ |
| | SAM-Adam | $85.26_{(0.15)}$ | $0.46_{(0.007)}$ | $0.0228_{(0.002)}$ | $0.874_{(0.004)}$ |
| | bSAM (ours) | $\mathbf{88.72}_{(0.24)}$ | $\mathbf{0.34}_{(0.005)}$ | $\mathbf{0.0163}_{(0.002)}$ | $\mathbf{0.903}_{(0.003)}$ |
| ResNet-20-FRN CIFAR-100 | SGD | $55.82_{(0.97)}$ | $1.91_{(0.025)}$ | $0.1695_{(0.005)}$ | $0.811_{(0.004)}$ |
| | SAM-SGD | $58.58_{(0.59)}$ | $1.60_{(0.022)}$ | $0.0989_{(0.005)}$ | $0.827_{(0.003)}$ |
| | SWAG | $56.53_{(0.40)}$ | $1.86_{(0.018)}$ | $0.1604_{(0.004)}$ | $0.814_{(0.004)}$ |
| | VOGN | $59.83_{(0.75)}$ | $1.44_{(0.019)}$ | $0.0756_{(0.005)}$ | $0.830_{(0.002)}$ |
| | Adam | $39.73_{(0.97)}$ | $2.29_{(0.045)}$ | $\mathbf{0.0295}_{(0.018)}$ | $0.775_{(0.004)}$ |
| | SAM-Adam | $53.25_{(0.80)}$ | $1.71_{(0.035)}$ | $0.0401_{(0.005)}$ | $0.818_{(0.005)}$ |
| | bSAM (ours) | $\mathbf{62.64}_{(0.33)}$ | $\mathbf{1.32}_{(0.001)}$ | $0.0311_{(0.003)}$ | $\mathbf{0.841}_{(0.004)}$ |

Table 1: Comparison without data augmentation. The results are averaged over 5 random seeds, with standard deviation shown in the brackets. The best statistical significant performance is shown in bold. bSAM (gray row) gives either comparable or better performance than the rest. Small scale experiments on MNIST are in Table 6 in App. F.8.

## 4 NUMERICAL EXPERIMENTS

In this section, we present numerical results and show that bSAM brings the best of the two worlds together. It gives an improved uncertainty estimate similarly to the best Bayesian approaches, but also improves test accuracy, just like SAM. We compare performance to many methods from the deep learning (DL) and Bayesian DL literature: SGD, Adam (Kingma & Ba, 2015), SWAG (Maddox et al., 2019), and VOGN (Osawa et al., 2019). We also compare with two SAM variants: SAM-SGD and SAM-Adam. Both are obtained by inserting the perturbed gradients into either SGD or Adam, as suggested in Foret et al. (2021); Bahri et al. (2022); Kwon et al. (2021). We compare these methods across three different neural network architectures and on five datasets of increasing complexity.

The comparison is carried out with respect to four different metrics evaluated on the validation set. The metrics are test accuracy, the negative log-likelihood (NLL), expected calibration error (ECE) (Guo et al., 2017) (using 20 bins) and area-under the ROC curves (AUROC). For an explanation of these metrics, we refer the reader to Osawa et al. (2019, Appendix H). For Bayesian methods, we report the performance of the predictive distribution $\widehat{p}(y \mid \mathcal{D}) = \int p(y \mid \mathcal{D}, \boldsymbol{\theta}) \, q(\boldsymbol{\theta}) \, \mathrm{d}\boldsymbol{\theta}$, which we approximate using an average over 32 models drawn from $q(\boldsymbol{\theta})$.

### 4.1 ILLUSTRATION ON A TOY EXAMPLE

Fig. 3 compares the predictive probability on a simple Bayesian logistic regression problem (Murphy, 2012, Ch. 8.4). For exact Bayes, we use numerical integration, which is feasible for this toy 2D problem. For the rest, we use diagonal Gaussians. White areas indicate probabilities around

| Model / Dataset | Method | Accuracy ↑ (higher is better) | NLL ↓ (lower is better) | ECE ↓ (lower is better) | AUROC ↑ (higher is better) |
|---|---|---|---|---|---|
| ResNet-20-FRN / CIFAR-10 | SGD | $91.68_{(0.26)}$ | $0.29_{(0.008)}$ | $0.0397_{(0.002)}$ | $0.915_{(0.002)}$ |
| | SAM-SGD | $\mathbf{92.29}_{(0.39)}$ | $0.25_{(0.004)}$ | $0.0266_{(0.003)}$ | $0.920_{(0.003)}$ |
| | Adam | $89.97_{(0.27)}$ | $0.41_{(0.021)}$ | $0.0610_{(0.003)}$ | $0.900_{(0.003)}$ |
| | SAM-Adam | $91.57_{(0.21)}$ | $0.26_{(0.004)}$ | $0.0329_{(0.002)}$ | $0.918_{(0.001)}$ |
| | bSAM (ours) | $\mathbf{92.16}_{(0.16)}$ | $\mathbf{0.23}_{(0.003)}$ | $\mathbf{0.0057}_{(0.002)}$ | $\mathbf{0.925}_{(0.001)}$ |
| ResNet-20-FRN / CIFAR-100 | SGD | $66.48_{(0.10)}$ | $1.20_{(0.007)}$ | $0.0524_{(0.004)}$ | $0.846_{(0.002)}$ |
| | SAM-SGD | $67.27_{(0.22)}$ | $1.19_{(0.011)}$ | $0.0481_{(0.001)}$ | $0.848_{(0.002)}$ |
| | Adam | $61.76_{(0.67)}$ | $1.66_{(0.049)}$ | $0.1582_{(0.006)}$ | $0.826_{(0.003)}$ |
| | SAM-Adam | $65.34_{(0.32)}$ | $1.23_{(0.012)}$ | $\mathbf{0.0166}_{(0.003)}$ | $0.847_{(0.002)}$ |
| | bSAM (ours) | $\mathbf{68.22}_{(0.44)}$ | $\mathbf{1.10}_{(0.013)}$ | $0.0258_{(0.003)}$ | $\mathbf{0.857}_{(0.004)}$ |
| ResNet-20-FRN / TinyImageNet | SGD | $52.01_{(0.36)}$ | $1.98_{(0.007)}$ | $0.0330_{(0.002)}$ | $\mathbf{0.832}_{(0.002)}$ |
| | SAM-SGD | $52.25_{(0.26)}$ | $1.97_{(0.013)}$ | $\mathbf{0.0155}_{(0.002)}$ | $0.827_{(0.005)}$ |
| | Adam | $49.04_{(0.38)}$ | $2.14_{(0.024)}$ | $0.0502_{(0.004)}$ | $0.820_{(0.004)}$ |
| | SAM-Adam | $51.17_{(0.45)}$ | $2.02_{(0.014)}$ | $0.0460_{(0.004)}$ | $0.828_{(0.004)}$ |
| | bSAM (ours) | $\mathbf{52.90}_{(0.35)}$ | $\mathbf{1.94}_{(0.009)}$ | $0.0199_{(0.003)}$ | $0.831_{(0.001)}$ |
| ResNet-18 / CIFAR-10 | SGD | $94.76_{(0.11)}$ | $0.21_{(0.006)}$ | $0.0304_{(0.001)}$ | $0.926_{(0.006)}$ |
| | SAM-SGD | $95.72_{(0.14)}$ | $0.14_{(0.004)}$ | $0.0134_{(0.001)}$ | $0.949_{(0.003)}$ |
| | bSAM (ours) | $\mathbf{96.15}_{(0.08)}$ | $\mathbf{0.12}_{(0.002)}$ | $\mathbf{0.0049}_{(0.001)}$ | $\mathbf{0.954}_{(0.001)}$ |
| ResNet-18 / CIFAR-100 | SGD | $76.54_{(0.26)}$ | $0.98_{(0.007)}$ | $0.0501_{(0.002)}$ | $0.869_{(0.003)}$ |
| | SAM-SGD | $78.74_{(0.19)}$ | $0.79_{(0.007)}$ | $0.0445_{(0.002)}$ | $0.887_{(0.003)}$ |
| | bSAM (ours) | $\mathbf{80.22}_{(0.28)}$ | $\mathbf{0.70}_{(0.008)}$ | $\mathbf{0.0311}_{(0.003)}$ | $\mathbf{0.892}_{(0.003)}$ |

Table 2: Comparison with data augmentation. Similar to Table 1, the shaded row show that bSAM consistently improves over the baselines and is the overall best method.

$0.5$, while red and blue are closer to $0$ and $1$ respectively. We see that the bSAM result is comparable to the exact posterior and corrects the overconfident predictions of SAM. Performing a Laplace approximation around the SAM solution leads to underconfident predictions. The posteriors are visualized in Fig. 5(c) and other details are discussed in App. F.1. We show results on another small toy example ('two moons') in App. F.2.

We can also use this toy problem to see the effect of using $f$ vs $f^{**}$ in the Bayesian objective. Fig. 7 in App. F.3 shows a comparison of optimizing Eq. 11 to an optimization of the original Bayesian objective Eq. 2. For both, we use full covariance to avoid the inaccuracies arising due to a diagonal covariance assumption. We find that the posteriors are very similar for both $f$ and $f^{**}$, indicating that the gap introduced by the relaxation is also small.

## 4.2 REAL DATASETS

We first perform a set of experiments without any data augmentation, which helps us to quantify the improvements obtained by the regularization induced by our Bayesian approach. This is useful in applications where it is not easy to do data-augmentation (for example, tabular or time series data). All further details about hyperparameter settings and experiments are in App. G. The results are shown in Table 1, and the proposed bSAM method overall performs best with respect to test accuracy as well as uncertainty metrics. bSAM also works well when training smaller networks on MNIST-like datasets, see Table 6 in App. F.8.

In Table 2 we show results with data augmentations (random horizontal flipping and random cropping). We consider CIFAR-10, CIFAR-100 and TinyImageNet, and still find bSAM to perform favorably, although the margin of improvement is smaller with data augmentation. We note that, for SGD on CIFAR-10 in Table 2, the accuracy (91.68% accuracy) is slightly better than the 91.25% reported in the original ResNet-paper (He et al., 2016, Table 6), by using a similar ResNet-20 architecture with filter response normalization (FRN), which has 0.27 million parameters.

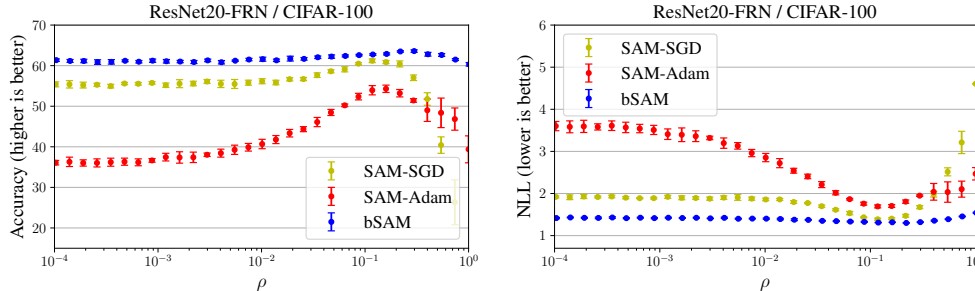

Figure 4: bSAM is less sensitive to the choice of $\rho$ than both SAM-SGD and SAM-Adam. We show accuracy and NLL. Plots for the other metrics are in App. F.4.

bSAM not only improves uncertainty estimates and test accuracy, it also reduces the sensitivity of SAM to the hyperparameter $\rho$. This is shown in Fig. 4 for SAM-SGD, SAM-Adam and bSAM, where bSAM shows the least sensitivity to the choice of $\rho$. Plots for other metrics are in App. F.4.

To understand the reasons behind the gains obtained with bSAM, we do several ablation studies. First, we study the effect of Gaussian sampling in line 3 in Alg. 1, which is not used in SAM. Experiments in App. F.5 show that Gaussian sampling consistently improves the test accuracy and uncertainty metrics. Second, in App. F.6, we study the effect of using posterior averaging over the quality of bSAM's predictions, and show that increasing the number of sampled models consistently improves the performance. Finally, we study the effect of "$m$-sharpness", where we distribute the mini-batch across several compute nodes and use a different perturbation for each node ($m$ refers to the number of splits). This has been shown to improve the performance of both SAM (Foret et al., 2021) and Bayesian methods (Osawa et al., 2019). In App. F.7, we show that larger values of $m$ lead to better performance for bSAM too. In our experiments, we used $m = 8$. For completeness, we show how $m$-sharpness is implemented in bSAM in App. E.4.

## 5 DISCUSSION

In this paper, we show a connection between SAM and Bayes through a Fenchel biconjugate. When the expected negative loss in the Bayes objective is replaced by the biconjugate, we obtain a relaxed Bayesian objective, which involves a maximum loss, similar to SAM. We showed that SAM can be seen as optimizing the mean while keeping variance fixed. We then derived an extension where the variances are optimized using a Bayesian learning algorithm applied to the relaxation. The numerical results show that the new variant improves both uncertainty estimates and generalization, as expected of a method that combines the strengths of Bayes and SAM.

We expect this result to be useful for researchers interested in designing new robust methods. The principles used in SAM are related to adversarial robustness which is different from Bayes. We show that the Fenchel conjugate is the bridge between the two, and we used it to obtain an extension of SAM. Since SAM gives an excellent performance on many deep-learning benchmark, the techniques used there will be useful in improving Bayesian methods. Finally, the use of convex duality can be useful to improve theoretical understanding of SAM-like methods. It is possible to use the results from convex-optimization literature to study and improve adversarial as well as Bayesian methods. Our work may open such directions and provide a new path for research in improving robustness. Expectations with respect to distributions arise naturally in many other problems, and often sampling is used to approximate them. Our paper gives a different way to approximate such expectations using convex optimization rather than sampling. Such problems may also benefit from our results.

ACKNOWLEDGEMENTS

We would like to thank Pierre Alquier for his valuable comments and suggestions. This work is supported by the Bayes duality project, JST CREST Grant Number JPMJCR2112.

AUTHOR CONTRIBUTIONS STATEMENT

List of Authors: Thomas Möllenhoff (T.M.), Mohammad Emtiyaz Khan (M.E.K.).

T.M. proposed the Fenchel biconjugates, their connections to SAM (Theorems 1 and 2), and derived the new bSAM algorithm. M.E.K. provided feedback to simplify these. T.M. conducted all the experiments with suggestions from M.E.K. Both authors wrote the paper together.

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

## APPENDIX

## A DERIVATION OF THE FENCHEL BICONJUGATE

Here, we give the derivation of the Fenchel Biconjugate given in Eq. 9. For an isotropic Gaussian, we have $\mathbf{T}(\boldsymbol{\theta}) = (\boldsymbol{\theta}, \boldsymbol{\theta}\boldsymbol{\theta}^\top)$ and $\boldsymbol{\lambda}' = (\mathbf{m}/\sigma^2, -1/(2\sigma^2)\mathbf{I})$. We will use the following identity to simplify,

$$\langle \boldsymbol{\lambda}', \mathbf{T}(\boldsymbol{\theta}) \rangle = \frac{\mathbf{m}^\top}{\sigma^2}\boldsymbol{\theta} + \mathrm{Tr}\left(-\frac{\mathbf{I}}{2\sigma^2}\boldsymbol{\theta}\boldsymbol{\theta}^\top\right) = -\frac{1}{2\sigma^2}\|\mathbf{m} - \boldsymbol{\theta}\|^2 + \frac{1}{2\sigma^2}\|\mathbf{m}\|^2. \tag{15}$$

Using this, the first term in Eq. 8 becomes

$$\langle \boldsymbol{\lambda}', \boldsymbol{\mu} \rangle = \mathbb{E}_{\boldsymbol{\theta} \sim q_\mu}[\langle \boldsymbol{\lambda}', \mathbf{T}(\boldsymbol{\theta}) \rangle] = \mathbb{E}_{\boldsymbol{\theta} \sim q_\mu}\left[-\frac{1}{2\sigma^2}\|\mathbf{m} - \boldsymbol{\theta}\|^2\right] + \frac{1}{2\sigma^2}\|\mathbf{m}\|^2. \tag{16}$$

Similarly, for the second term in Eq. 8, we use Theorem 1 and Eq. 15 to simplify,

$$\sup_{\boldsymbol{\mu}'' \in \mathcal{M}} \langle \boldsymbol{\mu}'', \boldsymbol{\lambda}' \rangle - \mathbb{E}_{\boldsymbol{\theta} \sim q_\mu}[-\ell(\boldsymbol{\theta})] = \sup_{\boldsymbol{\theta} \in \Theta} \langle \boldsymbol{\lambda}', \mathbf{T}(\boldsymbol{\theta}) \rangle + \ell(\boldsymbol{\theta})$$

$$= \sup_{\boldsymbol{\theta} \in \Theta} \left[-\frac{1}{2\sigma^2}\|\boldsymbol{\theta} - \mathbf{m}\|^2 + \frac{1}{2\sigma^2}\|\mathbf{m}\|^2 + \ell(\boldsymbol{\theta})\right]$$

$$= \left[\sup_{\boldsymbol{\epsilon}} \ell(\mathbf{m} + \boldsymbol{\epsilon}) - \frac{1}{2\sigma^2}\|\boldsymbol{\epsilon}\|^2\right] + \frac{1}{2\sigma^2}\|\mathbf{m}\|^2, \tag{17}$$

where in the last step we have performed a change of variables $\boldsymbol{\theta} - \mathbf{m} = \boldsymbol{\epsilon}$. Subtracting (17) from (16), we arrive at Eq. 9.

## B PROOF OF THEOREM 1

**Theorem 1.** *For a Gaussian distribution that takes the form* (5)*, the conjugate $f^*(\boldsymbol{\lambda}')$ can be obtained by taking the supremum over $\Theta$ instead of $\mathcal{M}$, that is,*

$$\sup_{\boldsymbol{\mu}'' \in \mathcal{M}} \langle \boldsymbol{\mu}'', \boldsymbol{\lambda}' \rangle - \mathbb{E}_{\boldsymbol{\theta} \sim q_{\mu''}}[-\ell(\boldsymbol{\theta})] = \sup_{\boldsymbol{\theta} \in \Theta} \langle \mathbf{T}(\boldsymbol{\theta}), \boldsymbol{\lambda}' \rangle - [-\ell(\boldsymbol{\theta})], \tag{10}$$

*assuming that $\ell(\boldsymbol{\theta})$ is lower-bounded, continuous and majorized by a quadratic function. Moreover, there exists $\boldsymbol{\lambda}'$ for which $f^*(\boldsymbol{\lambda}')$ is finite and, assuming $\ell(\boldsymbol{\theta})$ is coercive, $\mathrm{dom}(f^*) \subseteq \Omega$.*

*Proof.* We first pull out the expectation in the left-hand side of Eq. 10,

$$\sup_{\boldsymbol{\mu}'' \in \mathcal{M}} \langle \boldsymbol{\mu}'', \boldsymbol{\lambda}' \rangle - \mathbb{E}_{\boldsymbol{\theta} \sim q_{\mu''}}[-\ell(\boldsymbol{\theta})] = \sup_{\boldsymbol{\mu}'' \in \mathcal{M}} \mathbb{E}_{\boldsymbol{\theta} \sim q_{\mu''}}[\underbrace{\langle \boldsymbol{\lambda}', \mathrm{T}(\boldsymbol{\theta}) \rangle + \ell(\boldsymbol{\theta})}_{=J(\boldsymbol{\theta})}].$$

We will denote the expression inside the expectation by $J(\boldsymbol{\theta})$, and prove that the supremum over $\boldsymbol{\mu}''$ is equal to the supremum over $\boldsymbol{\theta}$. We will do so for two separate cases where the supremum $J_* = \sup_{\boldsymbol{\theta} \in \Theta} J(\boldsymbol{\theta}) < \infty$ is finite, and infinite respectively.

**When $J_*$ is finite:** For this case, our derivation proceeds in two steps, where we show that both the following two inequalities are true at the same time,

$$\sup_{\boldsymbol{\mu}'' \in \mathcal{M}} \mathbb{E}_{\boldsymbol{\theta} \sim q_{\mu''}}[J(\boldsymbol{\theta})] \leq J_*, \qquad \sup_{\boldsymbol{\mu}'' \in \mathcal{M}} \mathbb{E}_{\boldsymbol{\theta} \sim q_{\mu''}}[J(\boldsymbol{\theta})] \geq J_*.$$

This is only possible when the two are equal, which will establish the result.

To prove the first inequality, we take a sequence $\{\boldsymbol{\theta}_t\}_{t=1}^\infty$ for which $J(\boldsymbol{\theta}_t) \to J_*$ when sending $t \to \infty$. By the definition of supremum, there always exists such a sequence. From the sequence $\{\boldsymbol{\theta}_t\}_{t=1}^\infty$ we will now construct a sequence $\{\boldsymbol{\mu}_t''\}_{t=1}^\infty$ for which the expectation of $J(\boldsymbol{\theta})$ also converges to $J_*$. This will establish that the supremum over $\boldsymbol{\mu}''$ is larger or equal than the supremum over $\boldsymbol{\theta}$. It is only larger or equal, since hypothetically there could be another sequence of $\{\boldsymbol{\mu}_t''\}_{t=1}^\infty$ which achieves higher value. We will see soon that this cannot be the case.

We use the sequence $\{\boldsymbol{\mu}_t''\}_{t=1}^\infty$ where $\boldsymbol{\mu}_t'' = (\boldsymbol{\theta}_t, \boldsymbol{\Sigma}_t + \boldsymbol{\theta}_t \boldsymbol{\theta}_t^\top)$ with $\{\boldsymbol{\Sigma}_t\}_{t=1}^\infty \subset \mathbb{S}_+^{P \times P}$ is taken in the space of symmetric positive-definite matrices with the sequence such that $\boldsymbol{\Sigma}_t \to 0$. Now, we can see that the expectation $\mathbb{E}_{\boldsymbol{\theta} \sim q_{\boldsymbol{\mu}_t''}}[J(\boldsymbol{\theta})]$ converges to the supremum as follows:

$$
\begin{aligned}
\lim_{t \to \infty} \mathbb{E}_{\boldsymbol{\theta} \sim q_{\boldsymbol{\mu}_t''}}[J(\boldsymbol{\theta})] &= \lim_{t \to \infty} \mathbb{E}_{\boldsymbol{\epsilon} \sim \mathcal{N}(0, \mathbf{I})} \left[ J\left(\boldsymbol{\theta}_t + \boldsymbol{\Sigma}_t^{1/2} \boldsymbol{\epsilon}\right) \right] \\
&= \mathbb{E}_{\boldsymbol{\epsilon} \sim \mathcal{N}(0, \mathbf{I})} \left[ \lim_{t \to \infty} J\left(\boldsymbol{\theta}_t + \boldsymbol{\Sigma}_t^{1/2} \boldsymbol{\epsilon}\right) \right] \\
&= \sup_{\boldsymbol{\theta} \in \mathbb{R}^P} J(\boldsymbol{\theta}).
\end{aligned}
$$

In the above steps, we used the fact that $J$ is Gaussian-integrable and continuous. A sufficient condition for the integrability is that $\ell(\boldsymbol{\theta})$ is lower-bounded and majorized by quadratics. With this, we have established that the supremum over $\boldsymbol{\mu}''$ is at least as large as the one over $\boldsymbol{\theta}$.

To prove the second inequality, we will show that the supremum over $\boldsymbol{\mu}''$ is also less or equal than the supremum over $\boldsymbol{\theta}$. To see that, we note that $J(\boldsymbol{\theta}') \leq J_*$ for all $\boldsymbol{\theta}' \in \Theta$. This implies that for any distribution $q_{\boldsymbol{\mu}''}$, we have that $\mathbb{E}_{\boldsymbol{\theta}' \sim q_{\boldsymbol{\mu}''}}[J(\boldsymbol{\theta}')] \leq J_*$. Since the inequality holds for any distribution, we also have that $\sup_{\boldsymbol{\mu}'' \in \mathcal{M}} \mathbb{E}_{\boldsymbol{\theta}' \sim q_{\boldsymbol{\mu}''}}[J(\boldsymbol{\theta}')] \leq J_*$. Having established both inequalities, we now know that the suprema in (10) agrees whenever the supremum over $\boldsymbol{\theta}$ is finite.

**When $J_*$ is infinite:** For such cases, we will show that $\sup_{\boldsymbol{\mu}'' \in \mathcal{M}} \mathbb{E}_{\boldsymbol{\theta} \sim q_{\boldsymbol{\mu}''}}[J(\boldsymbol{\theta})] = \infty$, which will complete the proof. Again, by definition of supremum, there exists a sequence $\{\boldsymbol{\theta}_t\}_{t=1}^\infty$ for which $J(\boldsymbol{\theta}_t) \to \infty$ when $t \to \infty$. This means that for any $M > 0$ there is a parameter $\boldsymbol{\theta}_{t(M)}$ in the sequence such that $J(\boldsymbol{\theta}_{t(M)}) > M$. Now from the previous part of the proof, we know that we can construct a sequence $q_{\boldsymbol{\mu}_t''}$ of Gaussians whose expectation converges to $J(\boldsymbol{\theta}_{t(M)})$. Since the above holds for any $M > 0$, the supremum over the expectation is also $+\infty$.

**Conditions for existence of $\boldsymbol{\lambda}'$ for which $J_*$ is finite:** The final step of our proof is to give the conditions when there exist at $\boldsymbol{\lambda}'$ for which $J_*$ is finite, and therefore the result is non-trivial.

Since $\ell(\boldsymbol{\theta})$ is majorized by a quadratic function, there exists a constant $c \in \mathbb{R}$, vector $\mathbf{b} \in \mathbb{R}^P$ and matrix $\mathbf{A} \in \mathbb{S}_+^{P \times P}$ such that the following bound holds:

$$
\ell(\boldsymbol{\theta}) \leq c + \mathbf{b}^\top \boldsymbol{\theta} + \tfrac{1}{2} \boldsymbol{\theta}^\top \mathbf{A} \boldsymbol{\theta}.
$$

Then for the candidate $\boldsymbol{\lambda}' = (-\mathbf{b}, -\tfrac{1}{2}\mathbf{A})$ we know that $J(\boldsymbol{\theta}) = \ell(\boldsymbol{\theta}) + \langle \boldsymbol{\lambda}', \mathbf{T}(\boldsymbol{\theta}) \rangle \leq c$ has a finite upper bound for any $\boldsymbol{\theta}$. Hence, the supremum is finite for this choice of $\boldsymbol{\lambda}' \in \Omega$.

**Domain of $f^*(\boldsymbol{\lambda}')$:** For a Gaussian distribution, valid $(\boldsymbol{\lambda}_1', \boldsymbol{\lambda}_2') \in \Omega$ satisfy the constraint $\boldsymbol{\lambda}_2' \prec 0$. We now show that if $\boldsymbol{\lambda}' \notin \Omega$, then $f^*(\boldsymbol{\lambda}') = +\infty$. Since $\boldsymbol{\lambda}_2' \nprec 0$, there exists a direction $\mathbf{d} \neq 0$ such that $\mathbf{d}^\top \boldsymbol{\lambda}_2' \mathbf{d} \geq 0$ and $\boldsymbol{\lambda}_1'^\top \mathbf{d} \geq 0$. Taking the sequence $\{\boldsymbol{\theta}_t\}_{t=0}^\infty$ with $\boldsymbol{\theta}_t = t \cdot \mathbf{d}$ we see

$$
J(\boldsymbol{\theta}_t) = t \boldsymbol{\lambda}_1'^\top \mathbf{d} + t^2 \mathbf{d}^\top \boldsymbol{\lambda}_2' \mathbf{d} + \ell(t\mathbf{d}) \geq \ell(t\mathbf{d}) \to \infty \text{ as } t \to \infty,
$$

since coercivity of $\ell(\boldsymbol{\theta})$ implies that $\ell(\boldsymbol{\theta}_t) \to \infty$ for any sequence $\{\boldsymbol{\theta}_t\}_{t=1}^\infty$ with $\|\boldsymbol{\theta}_t\| \to \infty$. $\qquad \square$

Some additional comments on the assumptions used in the theorem are in order.

1. Any reasonable loss function that one would want to minimize is lower-bounded, so this assumption is always satisfied in practice.

2. The existence of an upper-bounding quadratic holds for functions with Lipschitz continuous gradient, which is a standard assumption in optimization known as the descent lemma. In practice, our conjugates are also applicable to cases where this assumption is not satisfied. In such a case, one can consider the conjugate function only in a local neighbourhood around the current solution by restricting the set of admissible $\boldsymbol{\theta}$ in the supremum. It may not matter that the bounding quadratic will eventually intersect the loss far away outside of our current region of interest.

3. We assumed that $\ell(\boldsymbol{\theta})$ is continuous since virtually all loss functions used in deep learning are continuous. Interestingly, if $\ell(\boldsymbol{\theta})$ is not continuous, the conjugate function $f^*$ will involve its semicontinous relaxation.

4. The coercivity assumption on $\ell(\boldsymbol{\theta})$ is another standard assumption in optimization. For non-coercive losses, there could exist quadratic upper-bounds which are flat in some direction. This would lead to a dual variable $\boldsymbol{\lambda}'$ at the boundary of $\Omega$.

The above Theorem 1 also holds for other exponential family distributions such as scalar Bernoulli and Gamma distributions, but we do not give more details here because they are not relevant for connecting to SAM. There are also cases where the result does not hold. The simplest case is a Gaussian with fixed covariance. The above proof does not work there, because the closure (see e.g., Csiszár & Matúš (2005); Malagò et al. (2011)) of the family of distributions does not contain all Dirac delta distributions. Intuitively, the requirement is that we should be able to approach any delta as a limit of a sequence of members of the family, which excludes the fixed-covariance case.

## C DERIVATION OF THE EQUIVALENT RELAXED OBJECTIVE (12)

Inserting the biconjugate (9) into the relaxed Bayes objective (11) yields

$$
\sup_{\boldsymbol{\mu}\in\mathcal{M}}\sup_{\mathbf{m},\sigma}-\left[\sup_{\boldsymbol{\epsilon}}\ell(\mathbf{m}+\boldsymbol{\epsilon})-\frac{1}{2\sigma^2}\|\boldsymbol{\epsilon}\|^2\right]+\mathbb{E}_{\boldsymbol{\theta}\sim q_{\boldsymbol{\mu}}}\left[-\frac{1}{2\sigma^2}\|\boldsymbol{\theta}-\mathbf{m}\|^2+\log p(\boldsymbol{\theta})-\log q_{\boldsymbol{\mu}}(\boldsymbol{\theta})\right],
$$

where we expanded the KLD as $\mathbb{D}_{\mathrm{KL}}(q_{\boldsymbol{\mu}}(\boldsymbol{\theta})\,\|\,p(\boldsymbol{\theta}))=\mathbb{E}_{\boldsymbol{\theta}\sim q_{\boldsymbol{\mu}}}[\log q_{\boldsymbol{\mu}}(\boldsymbol{\theta})]-\mathbb{E}_{\boldsymbol{\theta}\sim q_{\boldsymbol{\mu}}}[\log p(\boldsymbol{\theta})]$.

Interchanging the order of maximization and noticing that the first term does not depend on $\boldsymbol{\mu}$, we can find a closed-form expression for the maximization in $\boldsymbol{\mu}$ as follows:

$$
\begin{aligned}
\sup_{\boldsymbol{\mu}\in\mathcal{M}}&\mathbb{E}_{\boldsymbol{\theta}\sim q_{\boldsymbol{\mu}}}\left[-\frac{1}{2\sigma^2}\|\boldsymbol{\theta}-\mathbf{m}\|^2+\log p(\boldsymbol{\theta})-\log q_{\boldsymbol{\mu}}(\boldsymbol{\theta})\right]\\
&=\sup_{\boldsymbol{\mu}\in\mathcal{M}}\mathbb{E}_{\boldsymbol{\theta}\sim q_{\boldsymbol{\mu}}}\left[\log\frac{1}{\mathcal{Z}}\exp\left(-\frac{1}{2\sigma^2}\|\boldsymbol{\theta}-\mathbf{m}\|^2\right)p(\boldsymbol{\theta})-\log q_{\boldsymbol{\mu}}(\boldsymbol{\theta})\right]+\log\mathcal{Z}\\
&=\log\mathcal{Z}-\inf_{\boldsymbol{\mu}\in\mathcal{M}}\mathbb{D}_{\mathrm{KL}}\left(q_{\boldsymbol{\mu}}(\boldsymbol{\theta})\,\|\,\frac{1}{\mathcal{Z}}\exp\left(-\frac{1}{2\sigma^2}\|\boldsymbol{\theta}-\mathbf{m}\|^2\right)p(\boldsymbol{\theta})\right)\\
&=\log\mathcal{Z}.
\end{aligned}
$$

In the last step, the KLD is attained at zero with $q_{\boldsymbol{\mu}}(\boldsymbol{\theta})=\frac{1}{\mathcal{Z}}\exp\left(-\frac{1}{2\sigma^2}\|\boldsymbol{\theta}-\mathbf{m}\|^2\right)p(\boldsymbol{\theta})$.

Finally, we compute $\mathcal{Z}$ as

$$
\begin{aligned}
\mathcal{Z}&=\int\exp\left(-\frac{1}{2\sigma^2}\|\boldsymbol{\theta}-\mathbf{m}\|^2\right)p(\boldsymbol{\theta})\,\mathrm{d}\boldsymbol{\theta}\\
&=(2\pi\sigma^2)^{P/2}\int(2\pi\sigma^2)^{-P/2}\exp\left(-\frac{1}{2\sigma^2}\|\boldsymbol{\theta}-\mathbf{m}\|^2\right)p(\boldsymbol{\theta})\,\mathrm{d}\boldsymbol{\theta}\\
&=(\sigma^2)^{P/2}(\sigma^2+1/\delta_0)^{-P/2}\exp\left(-\frac{\|\mathbf{m}\|^2}{2(\sigma^2+\frac{1}{\delta})}\right),
\end{aligned}
$$

where in the last step we inserted the normalization constant for the product of the two Gaussians $\mathcal{N}(\boldsymbol{\theta}\,|\,\mathbf{m},\sigma^2\mathbf{I})$ and $\mathcal{N}(\boldsymbol{\theta}\,|\,0,\frac{1}{\delta_0}\mathbf{I})$ as for example given in Rasmussen & Williams (2006, App. 2).

This gives the following expression

$$
\begin{aligned}
\log\mathcal{Z}&=\frac{P}{2}\log(\sigma^2)-\frac{P}{2}\log(\sigma^2+1/\delta_0)-\frac{1}{2(\sigma^2+\frac{1}{\delta_0})}\|\mathbf{m}\|^2\\
&=\frac{P}{2}\log(\sigma^2)+\frac{P}{2}\log(\delta')-\frac{\delta'}{2}\|\mathbf{m}\|^2,
\end{aligned}\tag{18}
$$

where $\delta'=1/(\sigma^2+1/\delta_0)$.

Switching to a minimization over $\mathbf{m},\sigma^2$ and inserting the result for the exact $\boldsymbol{\mu}$-solution (18) leads to the final objective

$$
\inf_{\mathbf{m},\sigma^2}\left[\sup_{\boldsymbol{\epsilon}}\ell(\mathbf{m}+\boldsymbol{\epsilon})-\frac{1}{2\sigma^2}\|\boldsymbol{\epsilon}\|^2\right]+\frac{\delta'}{2}\|\mathbf{m}\|^2-\frac{P}{2}\log(\sigma^2)-\frac{P}{2}\log(\delta'),
$$

which is the energy function in (12).

# D  PROOF OF THEOREM 2

**Theorem 2.** *For every $(\rho, \delta)$, there exist $(\sigma, \delta')$ such that*

$$\arg\min_{\boldsymbol{\theta} \in \Theta} \mathcal{E}_{SAM}(\boldsymbol{\theta}; \rho, \delta) = \arg\min_{\mathbf{m} \in \Theta} \mathcal{E}_{relaxed}(\mathbf{m}, \sigma; \delta'), \tag{13}$$

*assuming that the SAM-perturbation satisfies $\|\boldsymbol{\epsilon}\| = \rho$ at a stationary point.*

*Proof.* We will show that any stationary point of $\mathcal{E}_{\text{SAM}}$ is also a stationary point of $\mathcal{E}_{\text{relaxed}}$ when $\sigma$ and $\delta'$ are chosen appropriately. The correspondence between the stationary points follows directly from the equivalences between proximal and trust-region formulations (Parikh & Boyd, 2013, Sec. 3.4).

To show this, we first write the two objectives in terms of perturbation $\boldsymbol{\epsilon}'$ and $\boldsymbol{\epsilon}$ respectively,

$$\min_{\mathbf{m}} \ \sup_{\boldsymbol{\epsilon}'} \ \ell(\mathbf{m} + \boldsymbol{\epsilon}') + \frac{1}{2\sigma^2}\|\boldsymbol{\epsilon}'\|^2 + \frac{\delta'}{2}\|\mathbf{m}\|^2,$$

$$\min_{\boldsymbol{\theta}} \ \sup_{\boldsymbol{\epsilon}} \ \ell(\boldsymbol{\theta} + \boldsymbol{\epsilon}) + \mathbf{i}\{\|\boldsymbol{\epsilon}\| \leq \rho\} + \frac{\delta}{2}\|\boldsymbol{\theta}\|^2,$$

where $\mathbf{i}\{\|\cdot\| \leq \rho\}$ is the indicator function which is zero inside the $\ell_2$-ball of radius $\rho$ and $+\infty$ otherwise.

Since these are min-max problems, we need an appropriate notion of "stationary point", which we consider here to be a local Nash equilibirum (Jin et al., 2020, Prop. 3). At such points, the first derivatives in $\boldsymbol{\theta}$ and $\boldsymbol{\epsilon}'$ (or $\boldsymbol{\epsilon}$) vanish. Taking the derivatives and setting them to 0, we get the following conditions for the two problems,

$$\delta'\mathbf{m}_* = -\nabla\ell(\mathbf{m}_* + \boldsymbol{\epsilon}'_*), \qquad\qquad \boldsymbol{\epsilon}'_* = -\sigma^2 \nabla\ell(\mathbf{m}_* + \boldsymbol{\epsilon}'_*),$$

$$\delta\boldsymbol{\theta}_* = -\nabla\ell(\boldsymbol{\theta}_* + \boldsymbol{\epsilon}_*), \qquad\qquad \boldsymbol{\epsilon}_* = -\frac{\rho}{\mu}\nabla\ell(\boldsymbol{\theta}_* + \boldsymbol{\epsilon}_*),$$

for some constant multiplier $\mu > 0$. Here, we used the assumption that the constraint $\|\boldsymbol{\epsilon}_*\| \leq \rho$ is active at our optimal point. Since the constraint is active, the negative gradient is an element of the (non-trivial) normal cone, which gives us the multiplier $\mu$ in the second line above.

The two equations are structurally equivalent. Any pair $(\mathbf{m}_*, \boldsymbol{\epsilon}'_*)$ satisfying the first line also satisfies the second line for $\sigma^2 = \rho/\mu$ and $\delta' = \delta$. The opposite is also true when using the pair $(\boldsymbol{\theta}_*, \boldsymbol{\epsilon}_*)$ in the first line, which proves the theorem. We assume that the constraint in the perturbation is active, that is, $\|\boldsymbol{\epsilon}_*\| = \rho$. This assumption would be violated if there exists a local maximum within a $\rho$-ball around the parameter $\boldsymbol{\theta}_*$. However, such a local maximum is unlikely to exist since the parameter $\boldsymbol{\theta}_*$ is determined by minimization and tends to lie inside a flat minimum. $\qquad\square$

# E  DERIVATION OF THE bSAM ALGORITHM

In this section of the appendix, we show how the natural gradient descent update (14) leads us to the proposed bSAM algorithm shown in Alg. 1. As mentioned in the main text, for bSAM we consider a generalized setting where our Gaussian distribution $q_{\boldsymbol{\mu}}(\boldsymbol{\theta}) = \mathcal{N}(\boldsymbol{\theta} \mid \boldsymbol{\omega}, \mathbf{V})$ has diagonal covariance. This corresponds to the following setting:

$$\mathbf{T}(\boldsymbol{\theta}) = (\boldsymbol{\theta}, \boldsymbol{\theta}\boldsymbol{\theta}^\top), \quad \boldsymbol{\mu} = \left(\boldsymbol{\omega}, \boldsymbol{\omega}^2 + \text{diag}(\mathbf{s})^{-1}\right), \quad \boldsymbol{\lambda} = \left(\mathbf{s} \cdot \boldsymbol{\omega}, -\tfrac{1}{2}\text{diag}(\mathbf{s})\right). \tag{19}$$

Here, $\boldsymbol{\omega} \in \mathbb{R}^P$ is the mean and $\text{diag}(\mathbf{s}) = \mathbf{V}^{-1}$ denotes the entries of the inverse diagonal covariance matrix. All operations (squaring, multiplication) are performed entrywise.

## E.1  THE GRADIENT OF THE BICONJUGATE

First, we discuss how to compute that gradient of the Fenchel biconjugate, as the BLR update (14) requires it in every iteration. The diagonal Gaussian setup leads to a slightly generalized expression

for the Fenchel biconjugate function:

$$-f^{**}(\boldsymbol{\mu}) = \min_{\mathbf{m}\in\mathbb{R}^P, \mathbf{b}\in\mathbb{R}_+^P} \left[ \sup_{\boldsymbol{\epsilon}\in\mathbb{R}^P} \ell(\mathbf{m}+\boldsymbol{\epsilon}) - \frac{1}{2}\|\boldsymbol{\Sigma}^{-\frac{1}{2}}\boldsymbol{\epsilon}\|^2 \right] + \mathbb{E}_{\boldsymbol{\theta}\sim q_{\boldsymbol{\mu}}}\left[ \frac{1}{2}\|\boldsymbol{\Sigma}^{-\frac{1}{2}}(\boldsymbol{\theta}-\mathbf{m})\|^2 \right]$$

$$= \min_{\mathbf{m}\in\mathbb{R}^P, \mathbf{b}\in\mathbb{R}_+^P} \left[ \sup_{\boldsymbol{\epsilon}\in\mathbb{R}^P} \ell(\mathbf{m}+\boldsymbol{\epsilon}) - \frac{1}{2}\|\boldsymbol{\Sigma}^{-\frac{1}{2}}\boldsymbol{\epsilon}\|^2 \right] + \frac{1}{2}\|\boldsymbol{\Sigma}^{-\frac{1}{2}}(\boldsymbol{\omega}-\mathbf{m})\|^2 + \frac{1}{2}\operatorname{tr}(\boldsymbol{\Sigma}^{-1}\mathbf{V}), \quad (20)$$

where $\boldsymbol{\Sigma} = \operatorname{diag}(1/\mathbf{b})$ denotes the diagonal covariance matrix with $1/\mathbf{b}$ on its diagonal.

It follows from results in convex analysis, for instance from Rockafellar & Wets (1998, Proposition 11.3), that $\nabla f^{**}(\boldsymbol{\mu}) = (\boldsymbol{\Sigma}_*^{-1}\mathbf{m}_*, -\frac{1}{2}\mathbf{b}_*)$, that is, the gradient of the biconjugate can be constructed from the optimal solution $(\mathbf{m}_*, \mathbf{b}_*)$ of the optimization problem (20).

In (20) there are three optimization steps: over $\boldsymbol{\epsilon}, \mathbf{m}$, and $\mathbf{b}$. For the first two, we will make very similar approximation to what is used for SAM (Foret et al., 2021), and simply add an additional optimization over $\mathbf{b}$. To simplify computation, we will make an additional approximation. Instead of an iterative joint minimization in $\mathbf{m}$ and $\mathbf{b}$, we will perform a single block-coordinate descent step where we first minimize in $\mathbf{m}$ for fixed $\mathbf{b} = \mathbf{s}$ and then afterwards in $\mathbf{b}$ for fixed $\boldsymbol{\omega} = \mathbf{m}$. This will give rise to a simple algorithm that works well in practice.

**Optimization with respect to $\boldsymbol{\epsilon}$.** To simplify the supremum in $\boldsymbol{\epsilon}$ we will use SAM's technique of local linearization (Foret et al., 2021) with one difference. Instead of using the current solution to linearize, we will sample a random linearization point $\boldsymbol{\theta} \sim \mathcal{N}(\boldsymbol{\theta}\,|\,\boldsymbol{\omega}, \operatorname{diag}(\mathbf{s})^{-1})$. This is preferable from a Bayesian perspective and is also used in the other Bayesian variants (Khan et al., 2018). Linearization at the sampled $\boldsymbol{\theta}$ then gives us,

$$\ell(\mathbf{m}+\boldsymbol{\epsilon}) \approx \ell(\boldsymbol{\theta}) + \langle \nabla\ell(\boldsymbol{\theta}), \mathbf{m}+\boldsymbol{\epsilon}-\boldsymbol{\theta}\rangle.$$

Using this approximation in (20), we get a closed form solution for $\boldsymbol{\epsilon}_* = \boldsymbol{\Sigma}\nabla\ell(\boldsymbol{\theta})$.

**Optimization with respect to $\mathbf{m}$.** To get a closed-form solution for the optimization in $\mathbf{m}$, we again consider a linearized loss: $\ell(\mathbf{m}+\boldsymbol{\epsilon}) \approx \ell(\boldsymbol{\omega}+\boldsymbol{\epsilon}) + \langle\nabla\ell(\boldsymbol{\omega}+\boldsymbol{\epsilon}), \mathbf{m}-\boldsymbol{\omega}\rangle$. Moreover, we set $\boldsymbol{\epsilon} = \mathbf{V}\nabla\ell(\boldsymbol{\theta})$, corresponding to a fixed $\mathbf{b} = \mathbf{s}$. This reduces the optimization of (20) to

$$\mathbf{m}_* = \arg\min_{\mathbf{m}\in\mathbb{R}^P} \langle\nabla\ell(\boldsymbol{\omega}+\boldsymbol{\epsilon}), \mathbf{m}\rangle + \frac{1}{2}\|\boldsymbol{\Sigma}_*^{-1/2}(\mathbf{m}-\boldsymbol{\omega})\|^2, \quad (21)$$

which has closed form solution $\mathbf{m}_* = \boldsymbol{\omega} - \boldsymbol{\Sigma}_*\nabla\ell(\boldsymbol{\omega}+\boldsymbol{\epsilon})$ which we here leave to depend $\boldsymbol{\Sigma}_*$.

**Optimization with respect to $\boldsymbol{\Sigma}$ via $\mathbf{b}$.** Inserting $\boldsymbol{\epsilon}_*$ into (20) with the linearized loss, the minimization over $\mathbf{b}$ while fixing $\mathbf{m} = \boldsymbol{\omega}$ also has a simple closed-form solution as shown below,

$$\mathbf{b}_* = \arg\min_{\mathbf{b}\in\mathbb{R}_+^P} \|\boldsymbol{\Sigma}^{1/2}\nabla\ell(\boldsymbol{\theta})\|^2 + \sum_{i=1}^P \frac{\mathbf{b}_i}{\mathbf{s}_i} = \sqrt{\mathbf{s}\cdot\mathbf{g}^2} = |\mathbf{g}|\cdot\sqrt{\mathbf{s}}, \quad (22)$$

where $\boldsymbol{\theta} \sim \mathcal{N}(\boldsymbol{\theta}\,|\,\mathbf{m}, \operatorname{diag}(\mathbf{s})^{-1})$ and $\mathbf{g} = \nabla\ell(\boldsymbol{\theta})$.

### E.2 SIMPLIFYING THE NATURAL-GRADIENT UPDATES

Having a way to approximate the biconjugate-gradient, we now rewrite and simplify the BLR update (14). The derivations are largely similar to the ones used to derive the variational online Newton approaches, see Khan et al. (2017; 2018); Khan & Rue (2021).

Inserting the parametrization (19) and our expression $\nabla f^{**}(\boldsymbol{\mu}) = (\boldsymbol{\Sigma}_*^{-1}\mathbf{m}_*, -\frac{1}{2}\mathbf{b}_*)$ into the BLR update (14), we get the following updates

$$\mathbf{s}\cdot\boldsymbol{\omega} \leftarrow (1-\alpha)\mathbf{s}\cdot\boldsymbol{\omega} + \alpha\boldsymbol{\Sigma}_*^{-1}\mathbf{m}_*,$$
$$\mathbf{s} \leftarrow (1-\alpha)\mathbf{s} + \alpha(\mathbf{b}_* + \delta_0),$$

where we also used the prior $\boldsymbol{\lambda}_0 = (0, -\frac{1}{2}\delta_0\mathbf{I})$. By changing the order of the updates, we can write them in an equivalent form that resembles an adaptive gradient method:

$$\mathbf{s} \leftarrow (1-\alpha)\mathbf{s} + \alpha(\mathbf{b}_* + \delta_0), \quad (23)$$
$$\boldsymbol{\omega} \leftarrow \boldsymbol{\omega} - \alpha\mathbf{V}\left[\boldsymbol{\Sigma}_*^{-1}(\boldsymbol{\omega}-\mathbf{m}_*) + \delta\boldsymbol{\omega}\right]. \quad (24)$$

Now, inserting the solutions $\boldsymbol{\Sigma}_*^{-1}(\boldsymbol{\omega} - \mathbf{m}_*) = \nabla\ell(\boldsymbol{\theta} + \boldsymbol{\epsilon})$ and $\mathbf{b}_* = |\nabla\ell(\boldsymbol{\theta})| \cdot \sqrt{\mathbf{s}}$ from Eqs. 21 and 22 into these updates, we arrive at

$$\mathbf{s} \leftarrow (1 - \alpha)\mathbf{s} + \alpha\left(|\nabla\ell(\boldsymbol{\theta})| \cdot \sqrt{\mathbf{s}} + \delta_0\right), \tag{25}$$

$$\boldsymbol{\omega} \leftarrow \boldsymbol{\omega} - \alpha\mathbf{V}\left(\nabla\ell(\boldsymbol{\omega} + \boldsymbol{\epsilon}) + \delta_0\boldsymbol{\omega}\right), \tag{26}$$

where $\boldsymbol{\epsilon} = \mathbf{V}\nabla\ell(\boldsymbol{\theta})$ and $\boldsymbol{\theta} \sim \mathcal{N}(\boldsymbol{\theta}\,|\,\boldsymbol{\omega}, \mathbf{V})$.

### E.3 FURTHER MODIFICATIONS FOR LARGE-SCALE DEEP LEARNING

Now we will consider further modifications to the updates (25) – (26) to arrive at our bSAM method shown in Alg. 1. These are similar to the ones used for VOGN (Osawa et al., 2019).

First, we consider a stochastic approximation to deal with larger problems. To that end, we now briefly describe the general learning setup. The loss is then written as a sum of individual losses, $\ell(\boldsymbol{\theta}) = \sum_{i=1}^{N} \ell_i(\boldsymbol{\theta})$ where $N$ is the number of datapoints. The $N$ datapoints are disjointly partitioned into $K$ minibatches $\{B_i\}_{i=1}^{K}$ of $B$ examples each.

We then apply a stochastic variant to the bound $N\frac{1}{K}\sum_{i=1}^{K} f_i^{**}(\boldsymbol{\mu}) \leq f^{**}(\boldsymbol{\mu})$, where the individual functions are $f_i(\boldsymbol{\mu}) = \frac{1}{B}\sum_{j \in B_i} \mathbb{E}_{\boldsymbol{\theta} \sim q_{\boldsymbol{\mu}}}[-\ell_j(\boldsymbol{\theta})]$. Essentially, the variant computes the gradient on an incrementally sampled $f_i^{**}$ rather than considering the full dataset.

With $\mathbf{s} = N\hat{\mathbf{s}}$, $\delta_0 = N\delta$ this turns (25) – (26) into the following updates:

$$N\hat{\mathbf{s}} \leftarrow (1 - \alpha)N\hat{\mathbf{s}} + \alpha\left[N\delta + N\sqrt{N\hat{\mathbf{s}}} \cdot |\mathbf{g}|\right], \tag{27}$$

$$\boldsymbol{\omega} \leftarrow \boldsymbol{\omega} - \alpha\left[N\delta\boldsymbol{\omega} + N\mathbf{g}_\epsilon\right]/(N\hat{\mathbf{s}}), \tag{28}$$

where $\mathbf{g} = \frac{1}{B}\sum_{j \in \mathcal{M}} \nabla\ell_j(\boldsymbol{\theta})$, $\boldsymbol{\theta} \sim \mathcal{N}(\boldsymbol{\theta}\,|\,\boldsymbol{\omega}, \mathrm{diag}(\boldsymbol{\sigma}^2))$, $\boldsymbol{\sigma}^2 = (N\hat{\mathbf{s}})^{-1}$ and $\mathcal{M}$ denotes a randomly drawn minibatch. Moreover, we have $\mathbf{g}_\epsilon = \frac{1}{B}\sum_{j \in \mathcal{M}} \nabla\ell_j(\boldsymbol{\omega} + \boldsymbol{\epsilon})$ with $\boldsymbol{\epsilon} = \mathbf{g}/(N\hat{\mathbf{s}}) = \rho\mathbf{g}/\hat{\mathbf{s}}$ where we introduce the step-size $\rho > 0$ in the adversarial step to absorb the $1/N$ factor and to gain additional control over the perturbation strength.

Following the practical improvements of Osawa et al. (2019), we introduce a damping parameter $\gamma > 0$ and seperate step-size $\beta_2 > 0$ on the precision estimate, as well as an exponential moving average for the gradient $\beta_1 > 0$. Under these changes, and dividing out the $N$-factors in (27) – (28) we arrive at the following method:

$$\mathbf{g}_m \leftarrow \beta_1\mathbf{g}_m + (1 - \beta_1)\left(\delta\boldsymbol{\omega} + \mathbf{g}_\epsilon\right), \tag{29}$$

$$\hat{\mathbf{s}} \leftarrow \beta_2\hat{\mathbf{s}} + (1 - \beta_2)\left[\delta + \gamma + \sqrt{N\hat{\mathbf{s}}} \cdot |\mathbf{g}|\right], \tag{30}$$

$$\boldsymbol{\omega} \leftarrow \boldsymbol{\omega} - \alpha\mathbf{g}_m/\hat{\mathbf{s}}, \tag{31}$$

with $\mathbf{g}$ and $\mathbf{g}_\epsilon$ defined as above. In practice, the method performed better without the $\sqrt{N}$-factor in the $\hat{\mathbf{s}}$-update. Renaming $\hat{\mathbf{s}}$ as $\mathbf{s}$, the updates (29) – (31) are equivalent to bSAM (Alg. 1).

### E.4 BSAM ALGORITHM WITH $m$-SHARPNESS

The bSAM algorithm parallelizes well onto multiple accelerators. This is done by splitting up a minibatch into $m$ parts and computing independent perturbations for each part in parallel. The algorithm remains exactly the same as Alg. 1, with lines $3 - 6$ replaced by the following lines $1 - 8$:

**1:** Equally partition $\mathcal{M}$ into $\{\mathcal{M}_1, \ldots, \mathcal{M}_m\}$ of $B$ examples each
**2: for** $k = 1 \ldots m$ in parallel **do**
**3:**    $\boldsymbol{\theta}_k \leftarrow \boldsymbol{\omega} + \mathbf{e}_k, \mathbf{e}_k \sim \mathcal{N}(\mathbf{e}_k\,|\,0, \boldsymbol{\sigma}^2), \;\; \boldsymbol{\sigma}^2 \leftarrow \mathbf{1}/(N \cdot \mathbf{s})$
**4:**    $\mathbf{g}_k \leftarrow (1/B)\sum_{i \in \mathcal{M}_k} \nabla\ell_i(\boldsymbol{\theta}_k)$
**5:**    $\boldsymbol{\epsilon}_k \leftarrow \mathbf{g}_k/\mathbf{s}$
**6:**    $\mathbf{g}_{\epsilon,k} \leftarrow (1/B)\sum_{i \in \mathcal{M}_k} \nabla\ell_i(\boldsymbol{\omega} + \mathbf{e}_k)$
**7: end for**
**8:** $\mathbf{g}_\epsilon \leftarrow (1/m)\sum_{k=1}^{m} \mathbf{g}_{\epsilon,k}, \; \mathbf{g} \leftarrow (1/m)\sum_{k=1}^{m} \mathbf{g}_k$

## F    ADDITIONAL DETAILS AND EXPERIMENTS

### F.1    DETAILS ON THE LOGISTIC REGRESSION EXPERIMENT

Fig. 5(a) shows the binary classification data-set (red vs blue circles) we adopted from (Murphy, 2012, Ch. 8.4). The data can be linearly seperated by a linear classifier with two parameters whose decision boundary passes through zero. We show the decision boundaries obtained from the MAP solution and from samples of the exact Bayesian posterior in Fig. 5(a).

Fig. 5(b) shows the 2D probability density function of the exact Bayesian posterior over the two parameters, where the black cross indicates the mode (MAP solution), and the orange triangles the weight-samples corresponding to the classifiers shown in the left figure. In Fig. 5(c), we show the solutions found by SAM, bSAM and an approximate Bayesian posterior. On this example, the posterior approximation found by bSAM is similar to the Bayesian one. Performing a Laplace approximation at the SAM solution leads to the covariance shown by the green dashed ellipse. It overestimates the extend of the posterior as it is based only on local information.

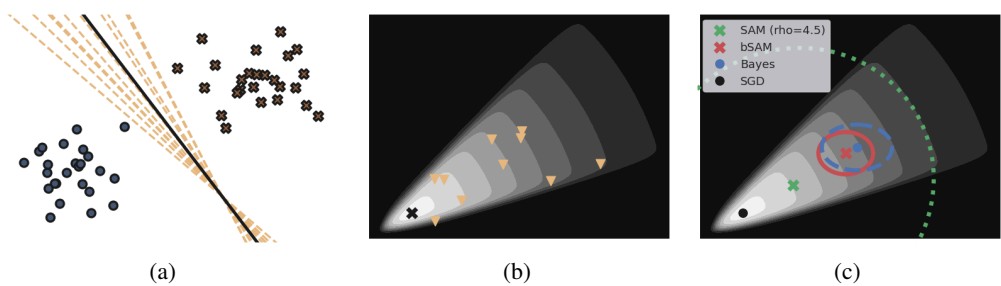

|  (a)  |  (b)  |  (c)  |

Figure 5: Posterior shapes for the logistic regression problem shown in Sec. 4.1.

### F.2    COMPARISON ON A LOW-DIMENSIONAL CLASSIFICATION PROBLEM

In Fig. 6, we compare predictive uncertainties of SAM and bSAM on the two-moons classification problem. We fit a small network $(2 - 24 - 12 - 12 - 1)$. For SAM+Laplace, we consider a diagonal GGN approximation to the Hessian (which was indefinite at the SAM solution) and use the linearized model for predictions, as suggested by Immer et al. (2021).

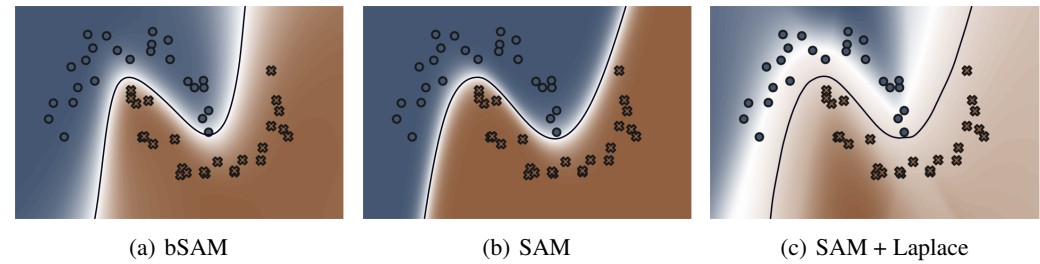

|  (a) bSAM  |  (b) SAM  |  (c) SAM + Laplace  |

Figure 6: bSAM improves SAM's predictive uncertainty. Using a Laplace approximation at the SAM solution can lead to an overestimation of the predictive uncertainty.

### F.3    EFFECT OF USING $f^{**}$ INSTEAD OF $f$

In this section, we empirically check the effect of using $f^{**}$ in the Bayesian objective instead of $f$. To that end, we compute an exact solution to the relaxed problem as well as the original problem (2) for a full Gaussian variational family.

For small problems, gradients of the biconjugate function can be easily approximated using convex optimization. To see this, notice that:

$$\nabla f^{**}(\boldsymbol{\mu}) = \arg\min_{\boldsymbol{\lambda}' \in \Omega} f^*(\boldsymbol{\lambda}') - \langle \boldsymbol{\lambda}', \boldsymbol{\mu} \rangle$$

$$= \arg\min_{\boldsymbol{\lambda}' \in \Omega, c \in \mathbb{R}} c - \langle \boldsymbol{\lambda}', \boldsymbol{\mu} \rangle \quad \text{s.t.} \quad c \geq f^*(\boldsymbol{\lambda}'). \tag{32}$$

This is a optimization problem with linear cost function and convex constraints. To implement the constraints, we approximate the loss by a maximum over linear functions, which is always possible for convex losses. The linear functions are first-order approximations of the loss at $L$ points $\boldsymbol{\theta}_i \sim q_{\boldsymbol{\mu}}$ drawn from the current variational distribution. Under this approximation, the constraint in (32) can be written as $L$ seperate constraints for $i \in \{1, \ldots, L\}$ as follows:

$$\sup_{\boldsymbol{\theta} \in \Theta} \boldsymbol{\theta}^\top \boldsymbol{\lambda}'_1 + \boldsymbol{\theta}^\top \boldsymbol{\lambda}'_2 \boldsymbol{\theta} + \ell(\boldsymbol{\theta}_i) + \langle \nabla\ell(\boldsymbol{\theta}_i), \boldsymbol{\theta} - \boldsymbol{\theta}_i \rangle \leq c$$

$$\Leftrightarrow -\frac{1}{4}(\boldsymbol{\lambda}'_1 + \nabla\ell(\boldsymbol{\theta}_i))^\top \boldsymbol{\lambda}'^{-1}_2 (\boldsymbol{\lambda}'_1 + \nabla\ell(\boldsymbol{\theta}_i)) - c \leq \langle \nabla\ell(\boldsymbol{\theta}_i), \boldsymbol{\theta}_i \rangle - \ell(\boldsymbol{\theta}_i). \tag{33}$$

The convex program (32) with convex constraints (33) is then solved using the software package CVXPY (Diamond & Boyd, 2016) to obtain the gradient of the biconjugate function.

Having the gradients of the biconjugate available, our posterior is obtained by the Bayesian learning rule of Khan & Rue (2021),

$$\boldsymbol{\lambda} \leftarrow (1 - \alpha)\boldsymbol{\lambda} + \alpha \begin{cases} \boldsymbol{\lambda}_0 + \nabla f(\boldsymbol{\mu}), & \text{for Bayes (2),} \\ \boldsymbol{\lambda}_0 + \nabla f^{**}(\boldsymbol{\mu}), & \text{for relaxed-Bayes (11).} \end{cases} \tag{34}$$

Fig. 7 compares both solutions obtained by iterating (34) on the logistic regression problem described in App. F.1. We can see that the relaxation induces a lower-bound as expected, but the relaxed solution is still a reasonable approximation to the true posterior.

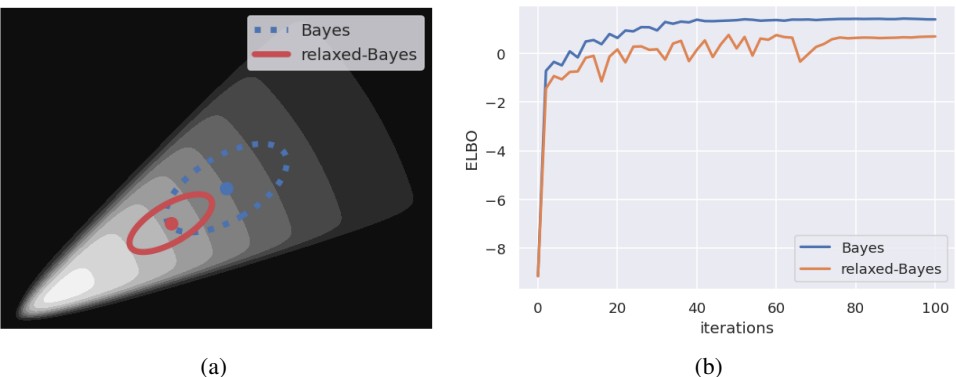

(a)                 (b)

Figure 7: Fig. 7(a) compares the posterior approximations obtained by solving (2) and our lower-bound (11) for a full Gaussian variational family. Fig. 7(b) shows the gap in the evidence-lower bound (ELBO) induced by the relaxation.

### F.4   SENSITIVITY OF BSAM TO THE CHOICE OF $\rho$

We show the sensitivity of all considered SAM variants to the hyperparameter $\rho$ in Fig. 8. As the bSAM algorithm learns the variance vector, the method is expected to be less sensitive to the choice of $\rho$. We confirm this here for a ResNet–20 on the CIFAR-100 dataset without data augmentation.

### F.5   ABLATION STUDY: EFFECT OF GAUSSIAN SAMPLING (LINE 3 IN ALG. 1)

In the derivation of the bSAM algorithm in Sec. 3 we performed a local linearization in order to approximate the inner problem. There, we claim that performing this local linearization not at the

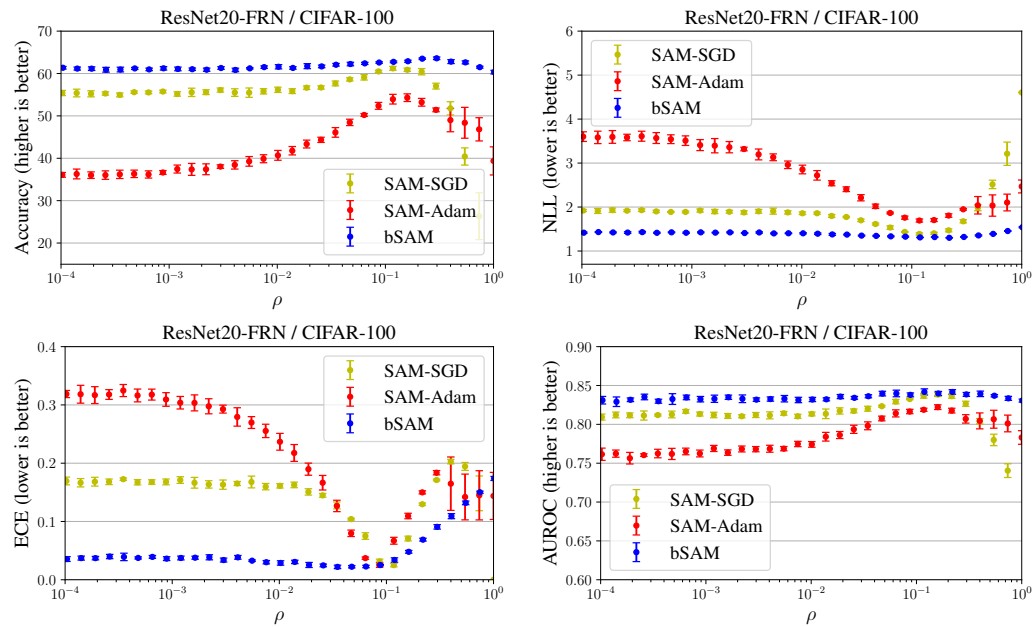

Figure 8: Sensitivity of SAM-variants to $\rho$. *For bSAM, we plot the markers at $100 \cdot \rho$ to roughly align the minima/maxima of the curves.* The proposed bSAM method adapts to shape of the loss and is overall more robust to misspecified parameter $\rho$ while also giving the overall best performance for all four metrics (Accuracy $\uparrow$, NLL $\downarrow$, ECE $\downarrow$, AUROC $\uparrow$).

mean but rather at a sampled point is preferable from a Bayesian viewpoint. A concurrent work by Liu et al. (2022) also shows that combining SAM with random sampling improves the performance. The following Table 3 confirms that the noisy linearization helps in practice.

| Model / Data | Method | Accuracy ↑ (higher is better) | NLL ↓ (lower is better) | ECE ↓ (lower is better) | AUROC ↑ (higher is better) |
|---|---|---|---|---|---|
| MLP / MNIST | bSAM w/o Noise | $98.67_{(0.04)}$ | $0.041_{(0.0011)}$ | $0.0030_{(0.0007)}$ | $0.981_{(0.001)}$ |
| | bSAM (Alg. 1) | $98.78_{(0.06)}$ | $0.038_{(0.0012)}$ | $0.0024_{(0.0004)}$ | $0.982_{(0.001)}$ |
| | **Improvement:** | **+00.11** | **+0.003** | **+0.0006** | **+0.001** |
| LeNet-5 / FMNIST | bSAM w/o Noise | $92.01_{(0.11)}$ | $0.23_{(0.002)}$ | $0.0128_{(0.0021)}$ | $0.918_{(0.002)}$ |
| | bSAM (Alg. 1) | $92.08_{(0.15)}$ | $0.22_{(0.005)}$ | $0.0066_{(0.0022)}$ | $0.920_{(0.003)}$ |
| | **Improvement:** | **+00.07** | **+0.01** | **+0.0062** | **+0.002** |
| ResNet-20-FRN / CIFAR-10 | bSAM w/o Noise | $87.74_{(0.28)}$ | $0.38_{(0.009)}$ | $0.0251_{(0.006)}$ | $0.896_{(0.004)}$ |
| | bSAM (Alg. 1) | $88.72_{(0.24)}$ | $0.34_{(0.005)}$ | $0.0163_{(0.002)}$ | $0.903_{(0.003)}$ |
| | **Improvement:** | **+00.98** | **+0.04** | **+0.0088** | **+0.007** |
| ResNet-20-FRN / CIFAR-100 | bSAM w/o Noise | $60.30_{(0.33)}$ | $1.44_{(0.009)}$ | $0.0226_{(0.003)}$ | $0.832_{(0.002)}$ |
| | bSAM (Alg. 1) | $62.64_{(0.34)}$ | $1.32_{(0.015)}$ | $0.0311_{(0.002)}$ | $0.841_{(0.003)}$ |
| | **Improvement:** | **+02.34** | **+0.12** | **−0.0085** | **+0.008** |

Table 3: Performing the loss-linearization at the noisy point rather than at the mean ("bSAM w/o Noise") typically improves the results with respect to all metrics.

## F.6    ABLATION STUDY: EFFECT OF BAYESIAN MODEL AVERAGING

In Table 4 we show how the predictive performance and uncertainty of bSAM depends on the number of MC samples used to approximate the integral in the Bayesian marginalization.

| Model / Data | Samples | Accuracy ↑ (higher is better) | NLL ↓ (lower is better) | ECE ↓ (lower is better) | AUROC ↑ (higher is better) |
|---|---|---|---|---|---|
| ResNet-20-FRN / CIFAR-10 | 0 (mean) | $88.34_{(0.22)}$ | $0.392_{(0.007)}$ | $0.0538_{(0.002)}$ | $0.9025_{(0.005)}$ |
| | 8 | $88.43_{(0.21)}$ | $0.349_{(0.006)}$ | $0.0212_{(0.002)}$ | $0.9038_{(0.005)}$ |
| | 64 | $88.59_{(0.20)}$ | $0.339_{(0.005)}$ | $0.0186_{(0.001)}$ | $0.9056_{(0.005)}$ |
| | 512 | $88.69_{(0.20)}$ | $0.338_{(0.005)}$ | $0.0173_{(0.002)}$ | $0.9047_{(0.005)}$ |
| | 2048 | $88.63_{(0.20)}$ | $0.338_{(0.005)}$ | $0.0173_{(0.001)}$ | $0.9058_{(0.005)}$ |

Table 4: Increasing the number of MC samples to approximate the marginal Bayesian predictive distribution tends to improve the performance in terms of NLL and ECE.

### F.7  ABLATION STUDY: DEPENDANCE OF SAM AND bSAM ON "m-SHARPNESS"

It has been observed in Foret et al. (2021) that the performance of SAM depends on the number $m$ of subbatches the minibatch is split into. This phenomenon is referred to as $m$-*sharpness*. From a theoretical viewpoint, a larger value of $m$ corresponds to a *looser* lower-bound for bSAM, as the sum of biconjugates is always smaller than the biconjugate of the sum.

Here we investigate the performance of SAM and bSAM on the parameter $m$ for a ResNet-20-FRN on CIFAR-100 without data augmentation. The other parameters are chosen as before, and the batch-size is set to $128$. The results are summarized in the following Table 5.

| Algorithm | $m$ | Accuracy ↑ (higher is better) | NLL ↓ (lower is better) | ECE ↓ (lower is better) | AUROC ↑ (higher is better) |
|---|---|---|---|---|---|
| bSAM | 1 | $61.38_{(0.27)}$ | $1.39_{(0.007)}$ | $0.0264_{(0.002)}$ | $0.836_{(0.003)}$ |
| | 2 | $61.47_{(0.46)}$ | $1.38_{(0.002)}$ | $0.0233_{(0.003)}$ | $0.837_{(0.003)}$ |
| | 4 | $62.35_{(0.32)}$ | $1.34_{(0.014)}$ | $\mathbf{0.0208}_{(0.003)}$ | $0.837_{(0.003)}$ |
| | 8 | $62.81_{(0.57)}$ | $1.31_{(0.017)}$ | $0.0327_{(0.004)}$ | $\mathbf{0.842}_{(0.003)}$ |
| | 16 | $\mathbf{63.31}_{(0.32)}$ | $\mathbf{1.30}_{(0.011)}$ | $0.0592_{(0.003)}$ | $0.840_{(0.003)}$ |
| SAM-SGD | 1 | $55.88_{(0.81)}$ | $1.82_{(0.032)}$ | $0.1501_{(0.004)}$ | $0.820_{(0.005)}$ |
| | 2 | $56.66_{(0.74)}$ | $1.76_{(0.041)}$ | $0.1421_{(0.006)}$ | $0.819_{(0.005)}$ |
| | 4 | $56.53_{(1.09)}$ | $1.73_{(0.058)}$ | $0.1289_{(0.006)}$ | $0.810_{(0.003)}$ |
| | 8 | $58.42_{(0.66)}$ | $1.60_{(0.030)}$ | $0.1017_{(0.005)}$ | $0.825_{(0.003)}$ |
| | 16 | $\mathbf{59.51}_{(0.67)}$ | $\mathbf{1.51}_{(0.040)}$ | $\mathbf{0.0681}_{(0.006)}$ | $\mathbf{0.829}_{(0.004)}$ |

Table 5: Effect of "$m$-sharpness" mini-batch size for SAM-SGD and bSAM for a ResNet-20-FRN on CIFAR-100 without data augmentation. As the effective minibatch size decreases, all performance metrics (accuracy and uncertainty) tend to improve. This boost in performance is not captured by our theory, and understanding it is an interesting direction for future work.

### F.8  EXPERIMENTS ON MNIST

In Table 6 we compare bSAM to several baselines for smaller neural networks on MNIST and FashionMNIST. Also in these smaller settings, bSAM performs competitively to state-of-the-art.

## G  CHOICE OF HYPERPARAMETERS AND OTHER DETAILS

### G.1  EXPERIMENTS IN TABLE 1 AND TABLE 6

In the following, we list the details of the experimental setup. First, we provide some general remarks, followed by tables of the detailed parameters used on each dataset.

For all experiments, the hyperparameters are selected using a grid-search over a moderate amount of configurations to find the best validation accuracy. Our neural network outputs the natural param-

| Model / Data | Method | Accuracy ↑ (higher is better) | NLL ↓ (lower is better) | ECE ↓ (lower is better) | AUROC ↑ (higher is better) |
|---|---|---|---|---|---|
| MLP MNIST | SGD | $98.63_{(0.06)}$ | $0.044_{(0.0012)}$ | $\mathbf{0.0028}_{(0.0007)}$ | $0.979_{(0.002)}$ |
| | SAM-SGD | $\mathbf{98.82}_{(0.03)}$ | $\mathbf{0.039}_{(0.0005)}$ | $0.0031_{(0.0002)}$ | $0.978_{(0.003)}$ |
| | SWAG | $98.65_{(0.04)}$ | $0.044_{(0.0002)}$ | $\mathbf{0.0028}_{(0.0005)}$ | $0.976_{(0.003)}$ |
| | VOGN | $98.54_{(0.03)}$ | $0.046_{(0.0008)}$ | $0.0033_{(0.0007)}$ | $\mathbf{0.980}_{(0.002)}$ |
| | Adam | $98.41_{(0.06)}$ | $0.050_{(0.0012)}$ | $0.0036_{(0.0007)}$ | $0.979_{(0.002)}$ |
| | SAM-Adam | $98.58_{(0.03)}$ | $0.046_{(0.0009)}$ | $0.0044_{(0.0005)}$ | $\mathbf{0.980}_{(0.002)}$ |
| | bSAM (ours) | $\mathbf{98.78}_{(0.06)}$ | $\mathbf{0.038}_{(0.0011)}$ | $\mathbf{0.0024}_{(0.0004)}$ | $\mathbf{0.982}_{(0.001)}$ |
| LeNet-5 FMNIST | SGD | $91.37_{(0.27)}$ | $0.32_{(0.007)}$ | $0.0429_{(0.0015)}$ | $0.897_{(0.004)}$ |
| | SAM-SGD | $\mathbf{91.95}_{(0.17)}$ | $\mathbf{0.22}_{(0.004)}$ | $\mathbf{0.0062}_{(0.0007)}$ | $\mathbf{0.917}_{(0.002)}$ |
| | SWAG | $91.38_{(0.27)}$ | $0.31_{(0.005)}$ | $0.0397_{(0.0026)}$ | $0.901_{(0.005)}$ |
| | VOGN | $91.24_{(0.25)}$ | $0.24_{(0.004)}$ | $\mathbf{0.0071}_{(0.0012)}$ | $\mathbf{0.916}_{(0.002)}$ |
| | Adam | $91.14_{(0.25)}$ | $0.33_{(0.005)}$ | $0.0450_{(0.0008)}$ | $0.897_{(0.005)}$ |
| | SAM-Adam | $91.66_{(0.19)}$ | $0.25_{(0.004)}$ | $0.0225_{(0.0026)}$ | $0.913_{(0.002)}$ |
| | bSAM (ours) | $\mathbf{92.10}_{(0.26)}$ | $\mathbf{0.22}_{(0.005)}$ | $\mathbf{0.0066}_{(0.0022)}$ | $\mathbf{0.920}_{(0.002)}$ |

Table 6: Comparison on small datasets.

eters of the categorical distribution as a minimal exponential family (number of classes minus one output neurons). The loss function is the negative log-likelihood.

We always use a batch-size of $B = 128$. For SAM-SGD, SAM-Adam and bSAM we split each minibatch into $m = 8$ subbatches, for VOGN we set $m = 16$ and consider independently computed perturbations for each subbatch. Finally, to demonstrate that bSAM does not require an excessive amount of tuning and parameters transfer well, we use the same hyper-parameters found on CIFAR-10 for the CIFAR-100 experiments.

We summarize all hyperparameters in Table 7.

**MNIST.** The architecture is a $784 - 500 - 300 - 9$ fully-connected multilayer perceptron (MLP) with ReLU activation functions. All methods are trained for 75 epochs. We use a cosine learning rate decay scheme, annealing the learning rate to zero. The hyperparameters of the individual methods are shown in Table 7. For SWAG, we run SGD for 60 epochs (with the same parameters). Then collect SWAG statistics with fixed $\alpha = 0.01$, $\beta_1 = 0.95$. For all SWAG runs, we set the rank to 20, and collect a sample every 100 steps for 15 epochs.

**FashionMNIST.** The architecture is a LeNet–5 with ReLU activations. We use a cosine learning rate decay scheme, annealing the learning rate to zero. We train all methods for 120 epochs. For SWAG, we run SGD for 105 epochs and collect 15 epochs with $\alpha = 0.001$ and $\beta_1 = 0.8$.

**CIFAR-10/100.** The architecture is a ResNet-20 with filter response normalization nonlinearity adopted from Izmailov et al. (2021). The ResNet-20 has the same number of parameters as the one used in the original publication (He et al., 2016). We train for 180 epochs and decay the learning rate by factor 10 at epoch 100 and 120. For SWAG, we run SGD for 165 epochs and collect for 15 epochs with $\alpha = 0.0001$, $\beta_1 = 0.9$.

## G.2 EXPERIMENTS IN TABLE 2

The bSAM method depends on the number of data-samples $N$. To account for the random cropping and horizontal flipping, we rescale this number by a factor 4 which improved the performance. This corresponds to a tempered posterior, as also suggested in Osawa et al. (2019).

In all experiments, all methods use cosine learning rate schedule which anneals the learning rate across 180 epochs to zero. The batch size is again chosen as $B = 128$.

For an overview over all the hyperparameters, please see Table 8.

| Configuration | Method | $\alpha$ | $\beta_1$ | $N \cdot \delta$ | $\rho$ | $\beta_2$ | $\gamma$ |
|---|---|---|---|---|---|---|---|
| MNIST / MLP | SGD | 0.1 | 0.95 | 24 | $\times$ | $\times$ | $\times$ |
| | VOGN | 0.0005 | 0.95 | 25 | $\times$ | 0.999 | 0.05 |
| | Adam | 0.0005 | 0.8 | 24 | $\times$ | $\bigcirc$ | $\bigcirc$ |
| | SAM-SGD | 0.1 | 0.95 | 30 | 0.05 | $\times$ | $\times$ |
| | SAM-Adam | 0.0005 | 0.8 | 24 | 0.02 | $\bigcirc$ | $\bigcirc$ |
| | bSAM | 0.1 | 0.8 | 15 | 0.01 | $\bigcirc$ | $\bigcirc$ |
| FashionMNIST / LeNet-5 | SGD | 0.01 | 0.8 | 60 | $\times$ | $\times$ | $\times$ |
| | VOGN | 0.0001 | 0.95 | 25 | $\times$ | 0.99 | 0.05 |
| | Adam | 0.001 | 0.9 | 60 | $\times$ | $\bigcirc$ | $\bigcirc$ |
| | SAM-SGD | 0.05 | 0.8 | 60 | 0.05 | $\times$ | $\times$ |
| | SAM-Adam | 0.001 | 0.9 | 60 | 0.02 | $\bigcirc$ | $\bigcirc$ |
| | bSAM | 0.1 | 0.95 | 50 | 0.002 | $\bigcirc$ | $\bigcirc$ |
| CIFAR-10 / ResNet-20-FRN | SGD | 0.1 | 0.9 | 25 | $\times$ | $\times$ | $\times$ |
| | VOGN | 0.001 | 0.9 | 25 | $\times$ | 0.999 | 0.01 |
| | Adam | 0.001 | 0.9 | 250 | $\times$ | $\bigcirc$ | $\bigcirc$ |
| | SAM-SGD | 0.05 | 0.9 | 25 | 0.05 | $\times$ | $\times$ |
| | SAM-Adam | 0.0003 | 0.9 | 25 | 0.1 | $\bigcirc$ | $\bigcirc$ |
| | bSAM | 0.1 | 0.9 | 25 | 0.001 | $\bigcirc$ | $\bigcirc$ |
| CIFAR-100 / ResNet-20-FRN | SGD | 0.1 | 0.9 | 30 | $\times$ | $\times$ | $\times$ |
| | VOGN | 0.001 | 0.9 | 25 | $\times$ | 0.999 | 0.01 |
| | Adam | 0.001 | 0.9 | 250 | $\times$ | $\bigcirc$ | $\bigcirc$ |
| | SAM-SGD | 0.05 | 0.9 | 25 | 0.05 | $\times$ | $\times$ |
| | SAM-Adam | 0.0003 | 0.9 | 25 | 0.1 | $\bigcirc$ | $\bigcirc$ |
| | bSAM | 0.1 | 0.9 | 25 | 0.001 | $\bigcirc$ | $\bigcirc$ |

Table 7: Hyperparameters used in the experiments without data augmentation. "$\times$" denotes that this method does not use that hyperparameter. "$\bigcirc$" indicates fixed hyperparameter across all datasets.

| Dataset | Method | $\alpha$ | $\beta_1$ | $N \cdot \delta$ | $\rho$ | $\beta_2$ | $\gamma$ |
|---|---|---|---|---|---|---|---|
| **CIFAR-10** | SGD | 0.1 | 0.95 | 10 | $\times$ | $\times$ | $\times$ |
| | Adam | 0.005 | 0.7 | 2.5 | $\times$ | $\bigcirc$ | $\bigcirc$ |
| | SAM-SGD | 0.03 | 0.95 | 25 | 0.01 | $\times$ | $\times$ |
| | SAM-Adam | 0.001 | 0.8 | 10 | 0.1 | $\bigcirc$ | $\bigcirc$ |
| | bSAM | 1.0 | 0.95 | 2 | 0.01 | $\bigcirc$ | $\bigcirc$ |
| **CIFAR-100** | SGD | 0.03 | 0.95 | 25 | $\times$ | $\times$ | $\times$ |
| | Adam | 0.005 | 0.9 | 1 | $\times$ | $\bigcirc$ | $\bigcirc$ |
| | SAM-SGD | 0.2 | 0.8 | 10 | 0.02 | $\times$ | $\times$ |
| | SAM-Adam | 0.005 | 0.7 | 1 | 0.2 | $\bigcirc$ | $\bigcirc$ |
| | bSAM | 1.0 | 0.95 | 2 | 0.01 | $\bigcirc$ | $\bigcirc$ |
| **TinyImageNet** | SGD | 0.1 | 0.95 | 20 | $\times$ | $\times$ | $\times$ |
| | Adam | 0.002 | 0.9 | 20 | $\times$ | $\bigcirc$ | $\bigcirc$ |
| | SAM-SGD | 0.1 | 0.95 | 50 | 0.01 | $\times$ | $\times$ |
| | SAM-Adam | 0.001 | 0.95 | 10 | 0.1 | $\bigcirc$ | $\bigcirc$ |
| | bSAM | 0.1 | 0.9 | 25 | 0.00005 | $\bigcirc$ | $\bigcirc$ |

Table 8: Hyperparameters used in the experiments with data augmentation shown in Table 2. "$\times$" denotes that this method does not use that hyperparameter. "$\bigcirc$" indicates fixed hyperparameter across all datasets.

### G.3 RESNET–18 EXPERIMENTS IN TABLE 2

We trained all methods over 180 epochs with a cosine learning rate scheduler, annealing the learning rate to zero. The batch-size is set to 200, and both SAM and bSAM use $m$-sharpness with $m = 8$.

| Dataset | Method | $\alpha$ | $\beta_1$ | $N \cdot \delta$ | $\rho$ | $\beta_2$ | $\gamma$ |
|---------|--------|----------|-----------|------------------|--------|-----------|----------|
| **CIFAR-10** | SGD | 0.03 | 0.9 | 25 | $\times$ | $\times$ | $\times$ |
| | SAM-SGD | 0.03 | 0.9 | 25 | 0.05 | $\times$ | $\times$ |
| | bSAM | 0.5 | 0.9 | 10 | 0.01 | $\bigcirc$ | $\bigcirc$ |
| **CIFAR-100** | SGD | 0.03 | 0.9 | 25 | $\times$ | $\times$ | $\times$ |
| | SAM-SGD | 0.03 | 0.9 | 25 | 0.05 | $\times$ | $\times$ |
| | bSAM | 0.5 | 0.9 | 10 | 0.01 | $\bigcirc$ | $\bigcirc$ |

Table 9: Hyperparameters used in the ResNet–18 experiments of Table 2. "$\times$" denotes that this method does not use that hyperparameter. "$\bigcirc$" indicates fixed hyperparameter across all datasets.

## H TABLE OF NOTATIONS

The following Table 10 provides an overview of the notations used throughout the paper. Multidimensional quantities are written in **bold-face**, scalars and scalar-valued functions are regular face.

| Symbol | Description |
|--------|-------------|
| $\langle \cdot, \cdot \rangle$ | An inner-product on a finite-dimensional vector-space. |
| $\| \cdot \|$ | The $\ell_2$-norm. |
| $f^*$, $f^{**}$ | Fenchel conjugate and biconjugate of the function $f$. |
| $\boldsymbol{\theta}$ | Parameter vector of the model, e.g., the neural network. |
| $\mathbf{T}(\boldsymbol{\theta})$ | Sufficient statistics of an exponential family distribution over the parameters. |
| $\boldsymbol{\mu}, \boldsymbol{\mu}', \boldsymbol{\mu}''$ | Expectation (or mean/moment) parameters of exponential family. |
| $\boldsymbol{\lambda}, \boldsymbol{\lambda}', \boldsymbol{\lambda}''$ | Natural parameters of an exponential family. |
| $\Omega$ | Space of valid natural parameters. |
| $\mathcal{M}$ | Space of valid expectation parameters. |
| $q_{\boldsymbol{\mu}}$ | Exponential family distribution with moments $\boldsymbol{\mu}$. |
| $q_{\boldsymbol{\lambda}}$ | Exponential family distribution with natural parameters $\boldsymbol{\lambda}$. |
| $A(\boldsymbol{\lambda})$ | Log-partition function of the exponential family in natural parameters. |
| $A^*(\boldsymbol{\mu})$ | Entropy of $q_{\boldsymbol{\mu}}$, the convex conjugate of the log-partition function. |
| $\boldsymbol{\omega}, v$ | Mean and variance of a normal distribution, corresponding to $\boldsymbol{\mu}$ and $\boldsymbol{\lambda}$. |
| $\mathbf{m}, \sigma^2$ | Mean and variance of another normal, corresponding to $\boldsymbol{\mu}'$ and $\boldsymbol{\lambda}'$. |
| $\mathcal{N}(\mathbf{e}|\mathbf{m}, \sigma^2 \mathbf{I})$ | $\mathbf{e}$ follows a normal distribution with mean $\mathbf{m}$ and covariance $\sigma^2 \mathbf{I}$. |
| $\delta$ | Precision of an isotropic Gaussian prior. |

Table 10: Notation.

