# OpenReview forum: "SAM as an Optimal Relaxation of Bayes"
_ICLR.cc/2023/Conference — ICLR 2023 notable top 5%_

### Official Review · Reviewer_Qr1q · 2022-10-19

**Confidence:** 3
**Correctness:** 4
**Technical Novelty And Significance:** 4
**Empirical Novelty And Significance:** 4
**Recommendation:** 8

**Clarity, Quality, Novelty And Reproducibility:**

I rate all four aspects as excellent: the write-up is of high quality and clarity, the theory and the proposed method are novel, and plenty of experimental details are provided in the appendix.

**Strength And Weaknesses:**

Strengths:
- A novel connection between two important ideas (SAM and Bayesian learning) that required a non-trivial amount of complex mathematical reasoning.
- A well-motivated, novel practical method driven by the theoretical results.
- Strong experimental results on a wide set of numerical benchmarks with a strong set of baseline methods.
- A clear, polished write-up with consistent notation. A precise comparison to the baseline SAM-Adam method with algorithmic differences highlighted line-by-line.
- Experiments to understand the sensitivity of the method to $\rho$, as well as additional ablation studies in the appendix.

Weaknesses:
- The toy experiment could be more informative: have authors considered doing non-linear classification with a NN instead? This might accentuate the differences between uncertainty estimates.
- I would recommend replacing the MNIST datasets in Table 1 with alternative, more challenging datasets. MNIST-based datasets are likely too easy for meaningful comparisons between methods, as indicated by more ambiguous results on these.
- A (brief) discussion on the computation cost of the method and how it compares to SAM/SAM-Adam would be useful.

**Summary Of The Paper:**

Sharpness-aware minimization (SAM) is a learning objective that improves upon maximum likelihood in terms of generalization, calibration and robustness of the resulting models. Intuitively, SAM drives the optimizer toward flatter minima, which are known to be desirable from the Bayes-optimal learning perspective. However, so far SAM has not been shown to directly correspond to any Bayesian learning objectives. In this paper authors derive this connection, showing that SAM is equivalent to minimizing a convex upper bound on a variational Bayesian learning objective. Authors then use this connection to propose an extension of SAM (Bayesian-SAM) that, in addition to the point-estimate of the parameters, estimates the parameter uncertainty, and leads to improved predictive performance and calibration in the numerical experiments.

**Summary Of The Review:**

I strongly recommend accepting this paper. It's a rare example of a paper that theoretically connects two important ideas, _and_ uses this connection to derive a novel practical method that works well on real data. The theoretical connection is non-trivial, and the method is presented and evaluated well (albeit the evaluation could be made even stronger, see my suggestions above).

---

> ### Author Response · Authors · 2022-11-16
> **Thank you for your review!**
>
> > The toy experiment could be more informative: have authors considered doing non-linear classification with a NN instead? This might accentuate the differences between uncertainty estimates.I would recommend replacing the MNIST datasets in Table 1 with alternative, more challenging datasets. MNIST-based datasets are likely too easy for meaningful comparisons between methods, as indicated by more ambiguous results on these.
>
> We have added an additional toy experiment on non-linear classification to Appendix I, and also new experiments in Appendix G.7. For the final version,we are hoping to include additional results on TinyImageNet and full ImageNet (ResNet-50). We agree with your comments about MNIST, but we added it since they are standard. We will consider moving it to the appendix for the final version and adding more challenging datasets.
>
> > A (brief) discussion on the computation cost of the method and how it compares to SAM/SAM-Adam would be useful.
>
> We discuss the computational overhead compared to SAM/SAM-Adam in the last paragraph of Section 3.

---

### Official Review · Reviewer_FsSv · 2022-10-24

**Confidence:** 5
**Correctness:** 4
**Technical Novelty And Significance:** 4
**Empirical Novelty And Significance:** 4
**Recommendation:** 8

**Clarity, Quality, Novelty And Reproducibility:**

As indicated the work was very clearly presented. Empirical findings were compelling and conducted on standard datasets using standard practices and architectures with ResNet sophistication or less. The precise algorithmic changes made to SAM+Adam were made abundantly clear in Algorithm 1.  Motivating biconjugation as an optimal convex relaxation of PAC-Bayes and for purpose of algorithmic development is, as far as I know, novel to that community.

A precise characterization of the algorithm with m-sharpness is missing; this reviewer kindly asks that it be added in at least the appendix.

**Details Of Ethics Concerns:**

Theory paper which attempts to explain and improve on existing work.

**Strength And Weaknesses:**

SAM is a SOTA optimizer which is poorly understood; any thoughtful story with appropriate empirical validation which explains how it works is a welcome contribution to the community. This paper easily exceeds that bar.  The mathematical details appeared correct and the prose was free from flaws or mistakes.

I only see one weakness in the paper which I share in the interest of constructive feedback and which in no way undermines my enthusiasm for the paper.

I would like to see much more study of the m-sharpness aspect of SAM, perhaps in a future work. M-sharpness was (seemingly?) only mentioned in the appendix. This is unfortunate because:

1. m-sharpness _further_ weakens the bound
2. m-sharpness leads to considerable test improvement.

The fact that weakening the bound improves results undermines the otherwise elegance of the paper's story. That is, we must now square the fact that an optimal convex relation of an otherwise defensible loss (von Neumann-Morgenstern and Savage utility theory / PAC-Bayes) is itself still somehow not optimal.  This is a hard pill to swallow because now that the loss is convex it is harder to lay blame to algorithmic concerns.

**Summary Of The Paper:**

This paper sheds light on the theoretical underpinnings of Sharpness Aware Minimization (SAM) by taking the biconjugation of the PAC-Bayes loss. The result is three-fold:

1. SAM can be understood as an optimal convex relaxation of the PAC-Bayesian loss.
2. This relaxation of the PAC-Bayes loss retains the logical sense sense in which it measures epistemic uncertainty. (The new loss being a looser bound.)
3. The PAC-Bayes relaxation is computed using convex optimization rather than Monte Carlo approximations.

Using the above insight, the authors modify SAM+Adam to fit a posterior mean and variance for Bayesian Neural Networks on standard image benchmarks.

**Summary Of The Review:**

This reviewer strongly endorses this paper be accepted. The authors present a simple and enlightening connection between (vanilla) SAM and Bayesian methods. We believe such connections are critical toward getting ever closer to understanding how/why modern DNN techniques perform so well and aligning those empirical successes with the theoretical underpinnings of Bayesian methods.

While we believe the lack of attention to m-sharpness was disappointing, we nevertheless recognize this work as a significant step toward better understand SAM and hopefully, ultimately leading to even more significant gains.

We thank the authors for their remarkably well written presentation and their resolve to offer insight into one of the recent empirical success stories, SAM.

---

> ### Author Response · Authors · 2022-11-16
> **Thank you for your review!**
>
> Thank you for the positive and encouraging evaluation of our work.
>
> > I would like to see much more study of the m-sharpness aspect of SAM, perhaps in a future work. M-sharpness was (seemingly?) only mentioned in the appendix. This is unfortunate because:
> m-sharpness further weakens the bound
> m-sharpness leads to considerable test improvement.
> [...]
> This is a hard pill to swallow because now that the loss is convex it is harder to lay blame to algorithmic concerns.
>
> We thank the reviewer for these interesting and detailed comments, and we fully agree. Paradoxically, tighter approximations of the generalization bound lead to worse results in practice. Currently, it remains difficult to optimize the loss faithfully despite convexity, especially for large neural networks. In future, we aim to explore whether more accurate approximations to the optimal convex lower bound give similar improvements as obtained by m-sharpness.
>
> We hope to add an additional discussion about this to the final version of the paper.
>
> > A precise characterization of the algorithm with m-sharpness is missing; this reviewer kindly asks that it be added in at least the appendix.
>
> We have added a description of the bSAM algorithm with m-sharpness to Appendix H.4.

---

### Official Review · Reviewer_VM1e · 2022-10-26

**Confidence:** 3
**Correctness:** 2
**Technical Novelty And Significance:** 3
**Empirical Novelty And Significance:** 2
**Recommendation:** 5

**Clarity, Quality, Novelty And Reproducibility:**

The paper is clear, given the complexity of the approach at hand, but some details are not given in enough detail.
Also, it would be better to:
- provide an explanation on the evaluation of the uncertainty estimate (maybe, I missed it in the text of the paper apart from ECE)
- direct link the baseline approach to the literature. Now the corresponding paragraph is vague and prevents exact attribution of e.g.
- The main Theorem 2 misses the list of assumptions used during the proof (the activity of the constraint, existence of the stationary point with zero derivatives given the hard constraint $i{\ldots}$. I assume that for the active constraint, the point would be a stationary one.

**Strength And Weaknesses:**

Strengths:
- Numerical experiments, to some extent, justify the method proposed in the paper.
- The idea is nice and can lead to some insights about how SAM works (but sadly, more on that is in the weakness part)

Weakness:
- Actually, losses (12) and (1) are very different from each other, as in (1) we take the supremum over a neighbourhood and (12) we look at all space of possible values $\varepsilon$. So, the related term $l(m + \varepsilon)$ in (12) can be arbitrarily larger than the corresponding term in (1), making the whole approach inspired by Fenchel biconjugates but not equivalent to it. We also note that in Fenchel biconjuates, we search for a convex analogue for the function we optimize. This makes little sense in general deep learning, as the loss is very far away from convex with many symmetries and local optima observed.
- So, we can't directly optimize (12).
- The accuracy of this approximation is, thus, should be the main concern in the paper. This is not true now. The experiments follow a common pipeline for DL papers comparing to other approaches and making reasonable ablation studies.
- what is the difference between ResNet-20-FRN and ResNet-20? The paper [1] reports similar results for ResNet-20 in Tables 2 and 3 for different variants of SAM for CIFAR10 and CIFAR100. For CIFAR10, they obtain 93.82% accuracy (c.t. bSAM 92.16% in the paper). For CIFAR100 they obtain 71.40% (c.t. bSAM 68.22% in the paper). What is the difference in the protocol that provides such a decrease in quality for your approach compared to the baselines?

Suggestions:
- I suppose that the biconjugate for the Exponential family is worth deeper examination, as it leads to convex problems, and thus, everything should be correct and can lead to interesting theoretical insights and analysis (also, I suppose that you can generalize Theorems to the paper in this more general case).


[1] Kwon, Jungmin, et al. "Asam: Adaptive sharpness-aware minimization for scale-invariant learning of deep neural networks." International Conference on Machine Learning. PMLR, 2021.


**Summary Of The Paper:**

The paper proposes a loss function that looks similar to SAM loss, but is a lower bound of Bayesian objective.
The authors state that the proposed pipeline can be used to obtain reasonable uncertainty estimates for NNs.

**Summary Of The Review:**

There is little empirical evidence that the proposed approach is better than other. It has only limited theoretical justification. So, I lean towards the reject the paper in its current state.

---

> ### Author Response · Authors · 2022-11-16
> **Thank you for your review!**
>
> > Actually, losses (12) and (1) are very different from each other, as in (1) we take the supremum over a neighbourhood and (12) we look at all space of possible values ε. So, the related term l(m+ε) in (12) can be arbitrarily larger than the corresponding term in (1), making the whole approach inspired by Fenchel biconjugates but not equivalent to it. We also note that in Fenchel biconjuates, we search for a convex analogue for the function we optimize. This makes little sense in general deep learning, as the loss is very far away from convex with many symmetries and local optima observed.
>
> There is a confusion here. We never claim that the losses in (12) and (1) are equivalent, rather only that their minimizers are equivalent (Theorem 2). We will rewrite the parts to avoid this confusion.
>
> We also disagree with the “convex analogue … making little sense in deep learning”. Our procedure does not convexify the original loss. The point is clearly explained in Fig. 2 and we request the reviewer to reconsider. The Fenchel bi-conjugates convexify the negative expected-loss in the mu-parameter (Fig. 2b), not the original loss in theta-parameter. As mentioned in the caption, “convexity of [the biconjugate] is lost” when we move to a different parameterization (Fig. 2c). But note that the biconjugate fully retains the original nonconvex-loss at the boundary of the parabola (shown with thick black curve in Fig. 2b). This is a subtle point, and it is due to the fact that any convex combination between points at the boundary of the parabola will not hit another point on the boundary of the parabola. This construction makes use of a (perhaps underappreciated) fact from convex analysis: convex functions can be nonconvex along the boundary of their domain, if the domain has appropriate shape!
>
> We hope that from this explanation, it becomes clear that our construction makes perfect sense for nonconvex loss functions.
>
> > what is the difference between ResNet-20-FRN and ResNet-20? The paper [1] reports similar results for ResNet-20 in Tables 2 and 3 for different variants of SAM for CIFAR10 and CIFAR100. For CIFAR10, they obtain 93.82% accuracy (c.t. bSAM 92.16% in the paper). For CIFAR100 they obtain 71.40% (c.t. bSAM 68.22% in the paper). What is the difference in the protocol that provides such a decrease in quality for your approach compared to the baselines?
>
> We follow the ResNet-20 proposed in the original ResNet-paper. The work [1] likely considers a “ResNet-20” that is different from the one described in the original paper of He et al. Such issues of “improper naming” are unfortunately common in the community and the details are described in this repository here:
> https://github.com/akamaster/pytorch_resnet_cifar10
>
> To be comparable to the original publication, we decided to use the original ResNet-20. Note that in the above repository (and in the original ResNet paper), the ResNet-20 reaches similar accuracy as reported in our paper.
>
> We have added to Appendix G.7 experiments on a larger ResNet-18 (a model intended for ImageNet but often applied to CIFAR-10; it has 11 million parameters).  For this model, SGD reaches 94.7%. This is improved by SAM to 95.7% and bSAM to 96.2%, see also the reply to reviewer Spsx. On CIFAR-100, bSAM improves SAM’s 78.7% to 80.2%.
>
> > provide an explanation on the evaluation of the uncertainty estimate (maybe, I missed it in the text of the paper apart from ECE)
>
> We clarified the uncertainty metrics at the beginning of Section 4 by referring more precisely to the literature.
>
> > direct link the baseline approach to the literature. Now the corresponding paragraph is vague and prevents exact attribution of e.g.
>
> We are not sure what exactly this comment is referring to. We provide citations to all the baseline methods we consider at the beginning of Section 4.
>
> > The main Theorem 2 misses the list of assumptions used during the proof (the activity of the constraint, existence of the stationary point with zero derivatives given the hard constraint.  I assume that for the active constraint, the point would be a stationary one.
>
> For our revised version, we have added the active constraint assumption to the statement of Theorem 2.
>
> > There is little empirical evidence that the proposed approach is better than other. It has only limited theoretical justification. So, I lean towards the reject the paper in its current state.
>
> We hope that our additional experiments in the new Appendix G.7 show more evidence of the practical relevance of the approach. Note that bSAM consistently outperforms all baselines with respect to all metrics.
>
> Moreover, we hope that our clarifications about the lifted Fenchel biconjugate make the reviewer understand that the approach is both theoretically sound and correct also for nonconvex deep learning scenarios.
>
> Given these clarifications and additional experiments, we kindly ask the reviewer to reevaluate their rating and consider increasing the score.

---

### Official Review · Reviewer_SPsx · 2022-10-26

**Confidence:** 3
**Correctness:** 3
**Technical Novelty And Significance:** 3
**Empirical Novelty And Significance:** 3
**Recommendation:** 6

**Clarity, Quality, Novelty And Reproducibility:**

This paper is well written in general, but clarity can be improved by simplifying the notation in Sec. 2, or perhaps including a notation table. The proposed method is novel and should be easy to reproduce.

In Eq. (4), f*(u) -> f(u).

**Strength And Weaknesses:**

Strength

Using the Fenchel biconjugate to bridge SAM and the Bayes objective is an interesting and insightful idea, and the resulting relaxation of the Bayes objective has shown good empirical performance. Moreover, the authors have conducted a variety of experiments and ablation studies on both toy and real datasets, providing deeper understanding of the proposed method.

Weaknesses
1. While the equivalence between SAM and the relaxed Bayes objective is interesting, it doesn't explain why the relaxed Bayes objective (or SAM) has better generalization performance than the original Bayes objective. In other words, it seems the "maximum loss" performs better than the "expected loss" despite the established connection, the reason of which is unclear.
2. The baselines used in the experiments are too weak. For instance, test accuracies around 95% and 75%, respectively, on CIFAR-10 and CIFAR-100 would be more convincing.
3. The readability of Sec. 2 needs improvement.

**Summary Of The Paper:**

The authors establish a connection between sharpness-aware minimization (SAM) and a Bayes objective by using a relaxation of the latter, such that the expected negative-loss is replaced by a tight convex lower bound, which is shown to be equivalent to SAM. Moreover, they propose a Bayesian extension of SAM that improves both generalization and uncertainty estimation.

**Summary Of The Review:**

This paper provides strong theoretical results, and extensive experimental results. However, some of the main questions remain unanswered, and some of the experimental results are not convincing enough due to weak baselines.

---

> ### Author Response · Authors · 2022-11-16
> **Thank you for your review.**
>
> > While the equivalence between SAM and the relaxed Bayes objective is interesting, it doesn't explain why the relaxed Bayes objective (or SAM) has better generalization performance than the original Bayes objective. In other words, it seems the "maximum loss" performs better than the "expected loss" despite the established connection, the reason of which is unclear.
>
> A complete understanding why SAM works better than expected loss is an open theoretical question in machine learning,and we indeed do not claim to fully answer it. We refer the reviewer to the last paragraph of Section 2, where we clearly mention that “the reasons behind the bad performance [of Bayes] remain unclear” and that we hope to use our connection in the future to address this problem.
>
> > The baselines used in the experiments are too weak. For instance, test accuracies around 95% and 75%, respectively, on CIFAR-10 and CIFAR-100 would be more convincing.
>
> We do not agree that the results are too weak. The baselines we used are similar to the other Bayesian deep learning papers, see, e.g., Table 3 in [2]. Nevertheless, to convince the reviewer, we have added additional experiments in Appendix G.7 where we use ResNet-18 architecture on CIFAR-10 and CIFAR-100 from the following repository: https://github.com/kuangliu/pytorch-cifar
>
> This ResNet-18 has more parameters (11M) compared to ResNet-20-FRN (200K) we used, but the latter is similar to the one used in the original ResNet publication of He et al. SGD on this larger ResNet-18 gives 94.7% on CIFAR-10 and 76.5% on CIFAR-100. Similar to other results, here too bSAM improves accuracy to 96.15% and 80.2%. See details in Appendix G.7. We are quite excited about these results, and hope that this is convincing for the reviewer too.
> For the final version, we may add a few more experiments involving ResNet-18 on TinyImageNet and a ResNet-50 on the full ImageNet dataset.
>
> [2] Izmailov et al., What Are Bayesian Neural Network Posteriors Really Like?, 2021.
>
>
> > The readability of Sec. 2 needs improvement. This paper is well written in general, but clarity can be improved by simplifying the notation in Sec. 2, or perhaps including a notation table. The proposed method is novel and should be easy to reproduce.
> ​​In Eq. (4), f*(u) -> f(u).
>
> Thanks for the typo! As per your suggestion, we have added a table of all considered notations to the Appendix J.

---

### Author Response · Authors · 2022-11-16
**Thank you for your reviews.**

We thank all reviewers for their careful reading and overall positive and encouraging evaluation of our work. We have uploaded a revised version, where the changes are highlighted in blue. To summarize our changes:

Based on the reviewer SPsx and VM1e’s feedback, we have added additional experiments in Appendix G.7 for a much larger ResNet on CIFAR-10 and CIFAR-100. For CIFAR-10, bSAM improves from SAM-SGD’s 95.7% to 96.15%, and for CIFAR-100 it improves from 78.74% to 80.22%, while simultaneously also giving much better calibration / uncertainty estimates.

We performed several minor changes requested by reviewers Qr1q and VM1e: experiments on a small nonlinear classification problem, table of notations, a missing assumption in one of the theorems, and addeding details about the evaluation metrics.

We hope that the revised paper addresses all concerns raised by the reviewers, and thank them again for their valuable feedback which helped us to improve the work.

---

### Public Comment · ~Kyunghun_Nam1 · 2023-06-19
**Inquiry Regarding Your Paper**

Dear Authors,

I hope this message finds you well. I recently had the opportunity to read your paper and found it profoundly impressive. As a researcher currently focusing on SAM, your work has been a significant source of inspiration for me.

I am reaching out to you today to seek clarification about a certain point in your work. I'm trying to determine whether a potential discrepancy I noticed is merely a typographical error or if it bears some specific meaning. In particular, the derivation of the relaxed-Bayes loss appears as "$\sup_{\epsilon^{\prime}} \ell( m + \epsilon^{\prime}) - \frac{1}{2 \sigma^2} \lVert \epsilon^{\prime} \rVert^2 + \frac{\delta^{\prime}}{2} \lVert m \rVert^2$", yet in the proof section for Theorem 2 in Appendix D, it is presented as "$\sup_{\epsilon^{\prime}} \ell(m + \epsilon^{\prime}) + \frac{1}{2 \sigma^2} \lVert \epsilon^{\prime} \rVert^2 + \frac{\delta^{\prime}}{2} \lVert m \rVert^2$". In other words, the negative sign has been replaced with a positive sign. Could you clarify whether this is a typo, or if there is a particular reason behind this change?

I would like to express my sincere gratitude once again for your exceptional research and your generous contribution to our shared field of study.

I eagerly look forward to your kind response.

Best Regards

---

> ### Author Response · Authors · 2023-06-19
> **Thank you for pointing out the typo!**
>
> Dear Kyunghun,
>
> This is indeed a typo which missed our attention. It should be with a minus rather than a plus in both equations.  We‘ll post an updated version on arXiv soon.
>
> Thanks!

---

> > ### Public Comment · ~Kyunghun_Nam1 · 2023-06-19
> > **Request for Additional Clarification Regarding Section 2**
> >
> > Dear Authors,
> >
> > I hope this message finds you in good health and high spirits. I am writing to express my gratitude for your prompt and thorough response to my previous query. Your explanations have been instrumental in furthering my understanding of your research.
> >
> > If it is the typo, then you should modify this part $\epsilon_{\star}^{\prime} = - \sigma^2 \nabla \ell(m_{\star} + \epsilon_{\star}^{\prime})$
> > to  $ + \sigma^2 \nabla \ell(m_{\star} + \epsilon_{\star}^{\prime})$ and thus the subsequent proof process.
> >
> > Building upon our ongoing dialogue, I am hoping to delve a little deeper into the specifics of your paper.
> > Specifically, I am interested in the transition from Equation $2$ ($L(q)$) to Equation $3$ ($\mathcal{E}_{Bayes}(m)$) in Section $2$. If possible, could you kindly provide a brief proof or explanation of this transition? It would be incredibly beneficial if you could consider adding this clarification either to an Appendix in the arXiv version of your paper or as a response to this inquiry.
> >
> > Your help in enhancing your research comprehension among fellow academics, such as myself, is immensely appreciated.
> >
> > I look forward to your kind response.

---

> > > ### Author Response · Authors · 2023-06-22
> > > **Thanks!**
> > >
> > > Dear Kyunghun,
> > >
> > > Yes, I agree with your suggested change to the proof -- we already modified the proof accordingly and will update the arXiv version soon :)
> > >
> > > About how to go from Eq. (2) to Eq. (3), I'm happy to explain it to you. It directly follows from the definition of KL-divergence between Gaussian distributions which is quite standard so we didn't include the derivation into the paper. If you send me a mail (thomas.moellenhoff@riken.jp) we could schedule a short discussion.

---

> > ### Public Comment · ~Kyunghun_Nam1 · 2023-06-22
> > **Further Inquiry Regarding the Connection Between SAM and Variational Inference**
> >
> > Dear Authors,
> >
> > I trust you are doing well and once again, I appreciate your patient responses to my queries. I apologize if my continued inquiries seem excessive, however, your research has greatly piqued my interest, and understanding it fully is of immense importance to my own work.
> >
> > Having read your paper, I am particularly intrigued by the relationship you've established between SAM and Bayes, particularly the use of the Fenchel biconjugate of the Evidence Lower Bound (ELBO), denoted as $\mathcal{L}(q)$ in Equation $2$, to derive another lower bound, the "relaxed-Bayes objective".
> >
> > As you're well aware, maximizing the ELBO is a common method in variational inference as it serves as a surrogate loss function to approximate complex posteriors using simpler, tractable distributions. In essence, by leveraging the Fenchel biconjugate, you've found the lower bound of the ELBO, which itself is a lower bound of the original objective function, thus establishing a structural identity between SAM and the relaxed-Bayes objective.
> >
> > Herein lies my question: Can we interpret the similarity between the original objective function and SAM in the same light? Given that you've uncovered a similarity between ELBO and SAM, and considering that the ELBO itself is a lower bound, is it feasible to consider replacing the maximization of ELBO with SAM in the context of variational inference if a similarity between the original objective function and SAM is established?
> >
> > Once again, I eagerly look forward to your response and deeply appreciate your time and attention in addressing these queries.
> >
> > Best Regards

---

### Decision · Program_Chairs · 2023-01-20

**Decision:**

Accept: notable-top-5%

**Justification For Why Not Higher Score:**

N/A

**Justification For Why Not Lower Score:**

The paper has solid theoretical and empirical results simultaneously. Their theoretical result, which relates SAM to an optimal relaxation of Bayes objective is very insightful and may open new doors to understanding how SAM can improve generalization. On the empirical side, the resulted algorithm can not only provide uncertainty estimates for predictions, but sometimes also outperform SAM in terms of test accuracy. The paper is very well written and the ideas well illustrated by multiple graphs.

**Metareview: Summary, Strengths And Weaknesses:**

This paper builds a beautiful connection between SAM and the Bayes objective, by showing that the former coincides with an optimal lower bound of the latter. The relaxation is constructed by replacing the expected negative-loss by its optimal convex lower bound using Fenchel biconjugate. As said by reviewer, and I echo that in strong agreement, "SAM is a SOTA optimizer which is poorly understood; any thoughtful story with appropriate empirical validation which explains how it works is a welcome contribution to the community". I believe this connection can open doors to better understanding the reasons behind SAM's success. Based on the proposed connection, the authors also introduce a modified version of SAM, namely bSAM, which can produce uncertainty estimates and also sometimes improve the accuracy of SAM. Two reviewers have rated the work 8 (accept) while the other two have rated borderline. I find the response to borderline reviews convincing though. In particular, in one case, the criticisms by the reviewer was due to a confusion about convexity claim which got clarified b the authors. Overall this paper has great contributions and interesting insights, and I recommend accept.



**Note From Pc:**

if the above contains the word "oral" or "spotlight" please see: "oral" presentation means -> notable-top-5% and "spotlight" means -> notable-top-25%. As stated in our emails, we are disassociating presentation type from AC recommendations